# An oxylipin signal confers protection against antifungal echinocandins in pathogenic aspergilli

Dante G. Calise [1,2], Sung Chul Park [1], Jin Woo Bok[1], Gustavo H. Goldman [3,4] & Nancy P. Keller [1,5] ✉

*Aspergillus fumigatus* is the leading causative agent of life-threatening invasive aspergillosis in immunocompromised individuals. One antifungal class used to treat *Aspergillus* infections is the fungistatic echinocandins, semisynthetic drugs derived from naturally occurring fungal lipopeptides. By inhibiting beta-1,3-glucan synthesis, echinocandins cause both fungistatic stunting of hyphal growth and repeated fungicidal lysis of apical tip compartments. Here, we uncover an endogenous mechanism of echinocandin tolerance in *A. fumigatus* whereby the inducible oxylipin signal 5,8-diHODE confers protection against tip lysis via the transcription factor ZfpA. Treatment of *A. fumigatus* with echinocandins induces 5,8-diHODE synthesis by the fungal oxygenase PpoA in a ZfpA dependent manner resulting in a positive feedback loop. This protective 5,8-diHODE/ZfpA signaling relay is conserved among diverse isolates of *A. fumigatus* and in two other *Aspergillus* pathogens. Our findings reveal an oxylipin-directed growth program—possibly arisen through natural encounters with native echinocandin producing fungi—that enables echinocandin tolerance in pathogenic aspergilli.

*Aspergillus fumigatus* is the leading etiological agent of a number of human diseases including the most severe pathology termed invasive aspergillosis (IA). Characterized by hyphal penetration of the lung endothelial layer, IA has an estimated mortality over fifty percent even with antifungal treatment[1]. Individuals with neutropenia such as hematopoietic stem cell transplant recipients comprise the historical host population, but the rise of the immunocompromised individual and COVID infections have resulted in expanding risk groups[2–4]. Treatment of IA is extremely limited with only three approved classes of antifungals. The polyene amphotericin B was the first approved antifungal to treat IA, but high rates of host toxicity have led to reduced use in current medical practice[5,6]. Triazole drugs targeting ergosterol synthesis are the recommended first-line treatment[6], but rising cases of azole resistance and poor clinical response necessitate

the use of salvage therapy with the cell wall targeting echinocandins[7–9]. Echinocandins are lipopeptide natural products synthesized by a subset of Leotiomycetes and Eurotiomycetes including some *Aspergillus* spp. but not *A. fumigatus*[10,11]. Semisynthetic derivatives of three natural echinocandins have been approved for clinical use in the treatment of fungal infections including IA[12].

Echinocandin drugs non-competitively inhibit the enzyme beta-1,3-glucan synthase, compromising the normal integrity of the fungal cell wall. Caspofungin, the most commonly used echinocandin, is described as fungistatic rather than fungicidal against aspergilli due to its inability to kill entire hyphae[13]. At the cellular level, however, the inhibitory effect of echinocandins is more complex. Treatment of *A. fumigatus* with caspofungin does in fact exert fungicidal activity on the apical most compartments of hyphae by lysing growing tips[14]. This is

[1]Department of Medical Microbiology and Immunology, University of Wisconsin–Madison, Madison, WI, USA. [2]Microbiology Doctoral Training Program, University of Wisconsin–Madison, Madison, WI, USA. [3]Faculdade de Ciências Farmacêuticas de Ribeirão Preto, Universidade de São Paulo, Ribeirão Preto, Brazil. [4]National Institute of Science and Technology in Human Pathogenic Fungi, Ribeirão Preto, Brazil. [5]Department of Plant Pathology, University of Wisconsin–Madison, Madison, WI, USA. ✉e-mail: npkeller@wisc.edu

due to the localization of beta-1,3-glucan synthase to the tips of hyphae where cell wall biosynthesis is occurring at the highest rate[15]. Importantly, this lytic effect of echinocandins is contained to only the tip compartment of growing hyphae by the plugging of septal pores. New hyphal tips can subsequently emerge behind blocked septa growing into or around lysed hyphal compartments allowing *A. fumigatus* hyphae to persist under treatment with these antifungals[16,17]. Nevertheless, these drugs are highly effective in restricting *A. fumigatus* growth due not only to repeated tip lysis but also to fungistatic stunting of hyphal expansion[14,15]. Focused studies using caspofungin show three characteristic morphological changes in treated hyphae: First, there is a shift from apical extension to lateral branching resulting in highly compact, hyperbranched microcolonies[15,17]. Second, an increase in septation results in a greater number of more closely spaced septa[15]. Third, the chitin content of the cell wall is greatly increased, compensating for impaired beta-1,3-glucan synthesis[18,19]. These developmental processes demonstrate a self-protective response of *A. fumigatus* to echinocandins.

The signals directing self-protective responses in microbes are poorly understood. Intriguingly, we recently found that treatment of *A. fumigatus* with its endogenous signaling oxylipin 5(S),8(R)-dihydroxyoctadeca-9Z,12Z-dienoic acid (5,8-diHODE) resulted in the same three morphological responses elicited by caspofungin but without the occurrence of tip lysis[20]. 5,8-diHODE is produced by the dihydroxylation of linoleic acid first to 8(R)-hydroxy-9Z,12Z-octadecadienoic acid (8-HODE) and subsequently to 5,8-diHODE by the linoleate diol synthase PpoA[21]. We found that *A. fumigatus* response to 5,8-diHODE requires the transcription factor ZfpA and that overexpression of ZfpA phenocopies treatment with 5,8-diHODE thus also resulting in hyperbranched, highly septate hyphae enriched in chitin[20]. Furthermore, deletion of ZfpA renders *A. fumigatus* more sensitive to caspofungin both in vitro and in vivo whereas its overexpression mediates protection against the antifungal drug[20,22,23]. Published RNA-sequencing data shows both the *ppoA* and *zfpA* transcripts are upregulated under caspofungin treatment[24]. Moreover, a study investigating the interactome of the septin AspB during caspofungin treatment found that the PpoA

enzyme physically associated with this cell wall associated protein only in the presence of the echinocandin drug[25].

Here we show that PpoA oxylipins act as signals directing a developmental program protective against echinocandins. We find that treatment of *A. fumigatus* with echinocandins induces *ppoA* expression and subsequent 8-HODE and 5,8-diHODE synthesis. Application of exogenous 5,8-diHODE protects *A. fumigatus* from fungicidal tip lysis by echinocandin antifungals. The oxylipin responsive transcription factor ZfpA is required for both induction of *ppoA* by caspofungin and for full protection by 5,8-diHODE against antifungal tip lysis. Protection by 5,8-diHODE against echinocandin tip lysis is also conserved in *A. flavus* and *A. nidulans* where homologs of ZfpA exhibit a conserved function. Our findings support a view that oxylipin signaling is an evolved protective mechanism in aspergilli during confrontations with exogenous echinocandin assault.

## Results

### Caspofungin induces production of the hyphal developmental signals 5,8-diHODE and 8-HODE

We recently showed that the fungal oxylipin 5,8-diHODE directs a morphological response in *A. fumigatus* characterized by hyperbranched, highly septate, chitin rich hyphae reminiscent of growth under treatment with the antifungal caspofungin[15,17,20]. Considering these reported similarities, we grew WT *A. fumigatus* Af293 in the presence of 5,8-diHODE, 8-HODE (the chemical precursor to 5,8-diHODE), caspofungin, or a vehicle control for 20 h before staining with calcofluor white in order to visualize the morphological responses to these compounds. In agreement with the aforementioned studies, all three chemical treatments resulted in similarly hyperbranched, highly septate hyphae rich in cell wall chitin (Fig. 1a). Furthermore, we found that the three treatments resulted in statistical increases in both lateral branching and cell wall chitin compared to vehicle controls (Supplementary Figs. 1 and 2).

The oxylipins 8-HODE and 5,8-diHODE are produced from linoleic acid by the fungal oxygenase PpoA[20]. A previous study showed that this enzyme co-localizes with the septin AspB under caspofungin

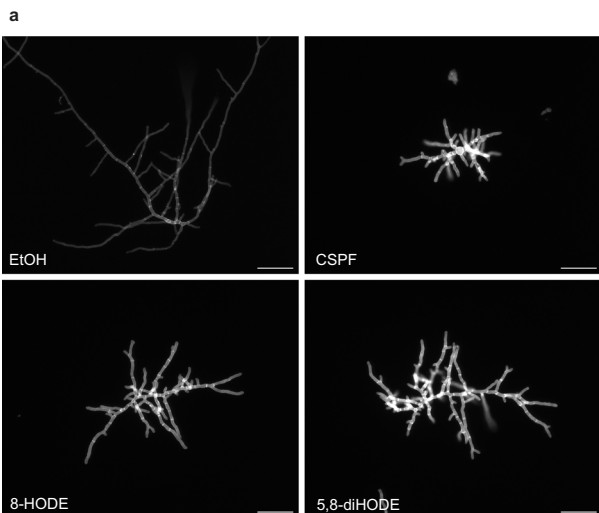

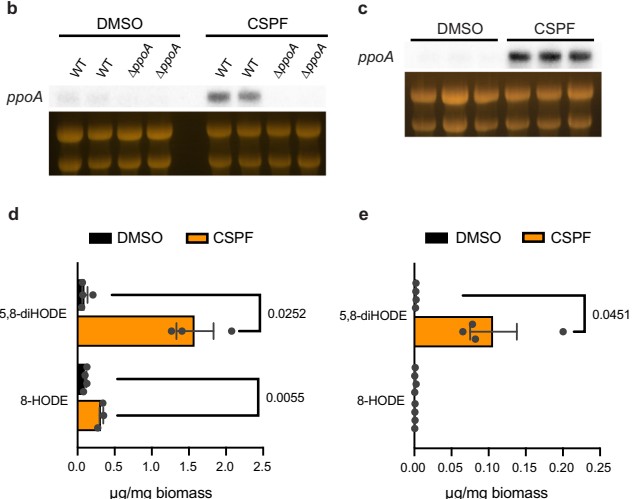

**Fig. 1 | Caspofungin treatment induces 5,8-diHODE and 8-HODE production by PpoA. a** Representative DAPI channel images of hyphae treated with 1% DMSO, 1 µg/mL caspofungin, 1 µg/mL 8-HODE, or 1 µg/mL 5,8-diHODE grown for 20 h is liquid GMM at 37 °C. Twenty-hour old hyphae grown in a cover glass bottom 96-well plate were stained with CFW and rinsed twice with sterile PBS before fluorescent imaging. Scale bars represent 50 microns. Micrographs are representative of three independent experiments. Northern blot analysis of *ppoA* expression in (**b**) WT Af293 and Δ*ppoA* strains or (**c**) WT CEA10 under treatment with 0.025% DMSO vehicle or 1 µg/mL caspofungin. Total RNA was extracted from replicate cultures

grown in liquid GMM at 250 RPM for 18.5 h before the addition of DMSO or CSPF for 90 min. Loading of 25 µg RNA per sample is shown by ethidium bromide staining of rRNA. 5,8-diHODE and 8-HODE per milligram of dry biomass extracted from (**d**) fungal tissue or (**e**) culture supernatant. WT *A. fumigatus* Af293 was grown at 37 °C and 250 RPM for 24 h in GMM plus 48 h more after the addition of 0.02% DMSO or 2 µg/mL CSPF. Oxylipins were extracted using mixed organic solvent and quantified on UHPLC-MS/MS by comparison to standard curves of purified oxylipin. Data points represent independent culture flasks (*n* = 3 or 4) and error bars denote SEM. *P* values shown were calculated using Welch's two-sided *t*-test.

exposure, suggesting that PpoA activity may be involved in cell wall remodeling processes during echinocandin stress[25]. Given this and the morphologically similar responses to these compounds, we hypothesized that caspofungin could induce *ppoA* expression and subsequent oxylipin synthesis. Northern blot analysis of WT *A. fumigatus* Af293 and a Δ*ppoA* mutant treated with caspofungin or a vehicle control revealed that caspofungin treatment robustly induced expression of the *ppoA* transcript in the WT strain (Fig. 1b). This induction of *ppoA* expression by caspofungin was also observed in another commonly used lab strain CEA10 (Fig. 1c). To test whether caspofungin induced the production of these oxylipins, we grew wild type *A. fumigatus* Af293 in the presence of caspofungin or a vehicle control and extracted oxylipins for quantification by UHPLC−HRMS/MS. We found that treatment with caspofungin resulted in significant increases in 5,8-diHODE detectable in both supernatant and tissue extracts as well as an increase in 8-HODE in fungal tissue (Fig. 1d, e). Production of another linoleic acid derived oxylipin 10-HODE by the enzyme PpoC was not affected suggesting that caspofungin specifically activates oxylipin production by PpoA (Supplementary Fig. 3).

## Cotreatment with 5,8-diHODE or 8-HODE protects *A. fumigatus* Af293 against caspofungin-mediated tip lysis

To visually assess the impact of oxylipin cotreatment on caspofungin efficacy, WT *A. fumigatus* Af293 conidia incubated with 1 µg/mL caspofungin, 10 µg/mL 5,8-diHODE, both caspofungin and 5,8-diHODE, or a vehicle control were observed by microscopic time-lapse imaging for 24 h at 37 °C. We found that cotreatment with 5,8-diHODE protected germlings against fungicidal lysis as observable by phase microscopy. Conidia under treatment with caspofungin alone often succumbed to fungicidal tip lysis at the germling stage with those that survived exhibiting the typical compact hyphal morphology associated with caspofungin. Cotreatment with 10 µg/mL 5,8-diHODE prevented tip lysis almost entirely, but hyphae still demonstrated the stunted hyperbranched growth characteristic of caspofungin treatment (Supplementary Movies 1–4). This suggests that 5,8-diHODE protects *A. fumigatus* against the fungicidal activity but not the fungistatic activity of caspofungin.

Next, we developed a microscopic assay to quantify germling survival under treatment with caspofungin or caspofungin/oxylipin cotreatment in which complete lysis of germlings was visualized by phase microscopy and confirmed by cessation of growth out to 24 h. We found that cotreatment with 10 µg/mL 5,8-diHODE, and to a lesser extent 8-HODE, significantly increased the survival of WT *A. fumigatus* Af293 hyphae after 16 h incubated with 1 µg/mL caspofungin (Fig. 2a, b). Furthermore, we found that 5,8-diHODE significantly reduced germling lysis in a concentration dependent manner with protection observed at concentrations as low as 0.01 µg/mL (Fig. 2c). Deletion of ppoA was previously shown to reduce the minimum effective concentration of caspofungin against *A. fumigatus*[25]. To test whether deletion or overexpression of the PpoA enzyme impacted susceptibility to caspofungin tip lysis, we performed germling lysis experiments with WT Af293, Δ*ppoA*, and OE::*ppoA* strains. While Δ*ppoA* germling survival was similar to wild type under treatment with caspofungin, OE::*ppoA* germling survival was significantly improved (Supplementary Fig. 4). This result supports a role for 5,8-diHODE in protection against caspofungin tip lysis but suggests that additional factors are also involved in echinocandin tolerance in *A. fumigatus*.

To determine if other non-PpoA synthesized oxylipins could also provide protection from caspofungin lysis, we compared tip lysis protection by 5,8-diHODE and 8-HODE to two structurally similar compounds, the *A. fumigatus* PpoC oxylipin 10-HODE and the mammalian oxylipin 9-HODE[26,27]. As before, 5,8-diHODE and 8-HODE cotreatments improved germling survival to nearly one hundred percent (Fig. 2d). Cotreatment with 10-HODE also improved germling survival but to a significantly lesser extent than either PpoA oxylipin,

and the mammalian oxylipin 9-HODE did not demonstrate any protective effect (Fig. 2d). Together these data suggest a specificity of PpoA oxylipin signaling in promoting tolerance of caspofungin in *A. fumigatus*.

## Protection by PpoA oxylipins against fungicidal activity of caspofungin is conserved among distinct isolates of *A. fumigatus*

Given the significant physiological heterogeneity among different isolates of *A. fumigatus*, including in their response to caspofungin[24,28,29], we sought to assess whether the protective effect of the PpoA oxylipins was conserved in other isolates of *A. fumigatus*. Time-lapse phase microscopy showed that germlings and developing hyphae of WT CEA10, another common lab strain, also appeared to be protected against caspofungin tip lysis by cotreatment with 5,8-diHODE. As with Af293, caspofungin treatment alone resulted in the fungicidal lysis of many germlings and circumscribed growth of the surviving hyphae whereas cotreatment with caspofungin and 5,8-diHODE drastically reduced tip lysis but still stunted hyphal growth (Supplementary Movies 5–8). We subsequently quantified WT CEA10 germling survival under treatment with caspofungin alone, 5,8-diHODE/caspofungin cotreatment, and 8-HODE/caspofungin cotreatment with the appropriate vehicle and oxylipin controls. We found that 5,8-diHODE and 8-HODE cotreatment both significantly improved survival of WT *A. fumigatus* CEA10 germlings after 16 h incubated with 2 µg/mL caspofungin (Fig. 2e, f). Similar protection by 5,8-diHODE and 8-HODE against caspofungin tip lysis was also observed in a third common lab strain of *A. fumigatus* AfS35 (Supplementary Fig. 5). Furthermore, the clinical isolate CM7555, previously shown to be highly tolerant to caspofungin[28], was assessed for germling lysis under 2 µg/mL caspofungin with or without 5,8-diHODE cotreatment relative to a CEA10 control. Compared to WT CEA10, CM7555 was less susceptible to lysis by caspofungin with a difference in survival of germlings of about 15% at 16 h. Despite differences in susceptibility to caspofungin, both strains showed close to one-hundred percent survival of germlings cotreated with 5,8-diHODE (Supplementary Fig. 6). Together these findings suggest that 5,8-diHODE is a conserved signal protective against caspofungin mediated hyphal damage in *A. fumigatus*.

## 5,8-diHODE is induced by and protective against treatment with micafungin

To determine whether induction of oxylipin synthesis by PpoA was limited to caspofungin or a more general response to antifungal stress, we extracted and quantified 5,8-diHODE and 8-HODE, as before, in wild-type *A. fumigatus* Af293 treated with caspofungin, micafungin, voriconazole, or a vehicle control. We found that treatment with micafungin produced a significant increase in 5,8-diHODE in fungal tissue comparable to those under caspofungin treatment. Conversely, voriconazole did not induce any detectable increase in 5,8-diHODE. We also observed a smaller but still detectable increase in 8-HODE in fungal tissue under micafungin but not voriconazole treatment (Fig. 3a). This finding suggests that 5,8-diHODE production is a specific response to the cell wall targeting echinocandins and not the membrane targeting azoles.

Considering this apparent conserved response to another echinocandin, we next tested whether 5,8-diHODE cotreatment protects WT *A. fumigatus* Af293 germlings against micafungin mediated lysis. As with caspofungin, we found that cotreatment with 10 µg/mL 5,8-diHODE and to a lesser extent 8-HODE significantly increased the survival of WT Af293 germlings after 16 h growth with 1 µg/mL micafungin (Fig. 3b). Similarly, cotreatment with 5,8-diHODE or 8-HODE protected WT CEA10 germlings against fungicidal lysis by micafungin (Fig. 3c). To assess whether 5,8-diHODE could protect against other cell wall perturbing agents, we grew WT *A. fumigatus* Af293 conidia under increasing concentrations of Congo red with or without a 10 µg/mL 5,8-diHODE cotreatment. Representative microscopic

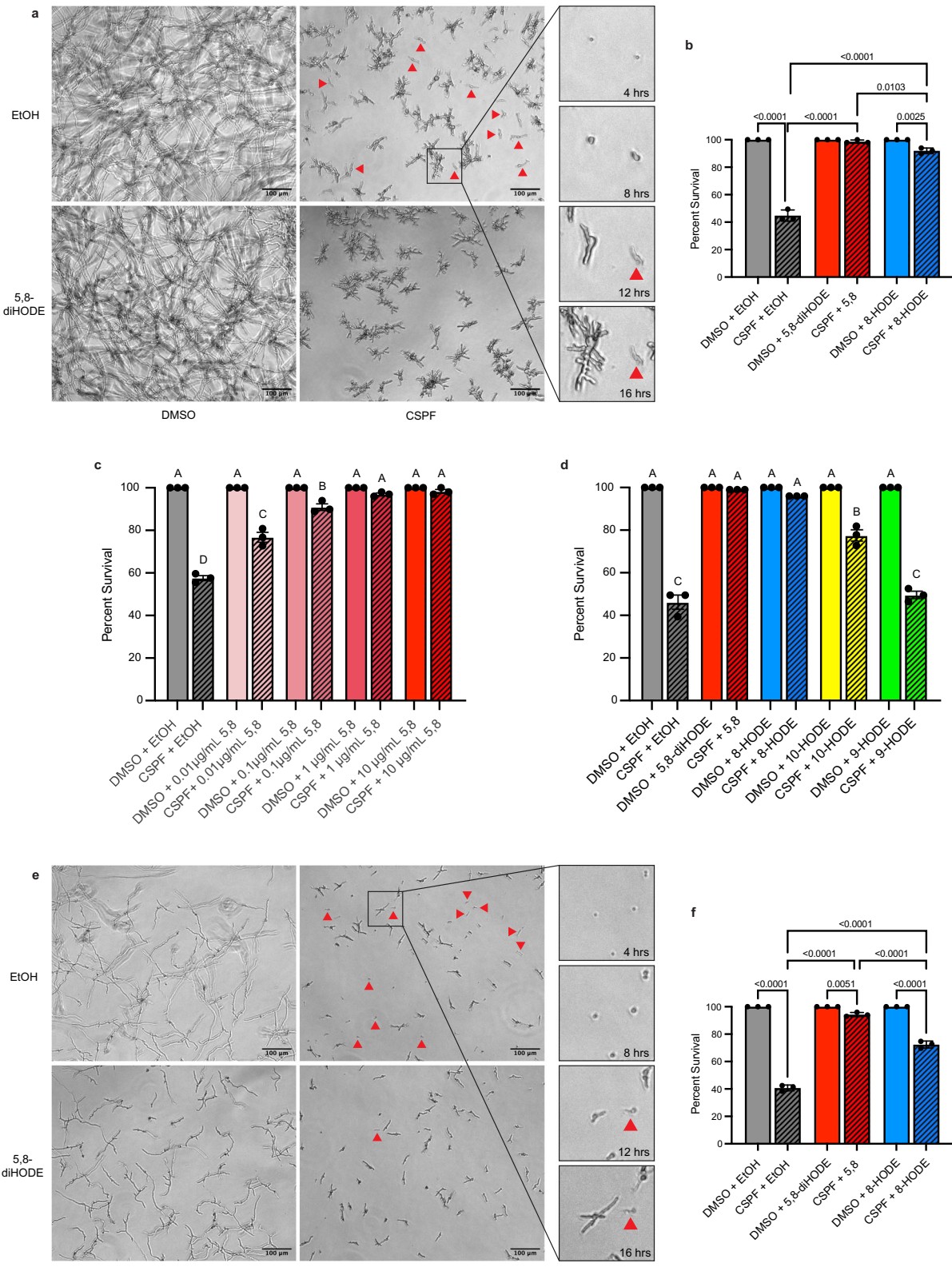

images of hyphal growth after 16 h showed that 5,8-diHODE was not protective against but rather synergized with Congo red causing the formation of swollen and lysed conidia at lower concentrations than without 5,8-diHODE cotreatment (Fig. 3d). These findings reveal that the induction of PpoA oxylipin production is a conserved, protective response specific to echinocandin stress.

## Protection by 5,8-diHODE against caspofungin tip lysis is conserved among pathogenic aspergilli

The role of PpoA has been studied in two other *Aspergillus* species, *A. flavus* and *A. nidulans*. *A. flavus* is the second most common human pathogenic *Aspergillus* spp[30]. while *A. nidulans* has emerged as a serious pathogen of chronic granulomatous disease

**Fig. 2 | Cotreatment with 5,8-diHODE or 8-HODE protects *A. fumigatus* against caspofungin-mediated tip lysis. a** WT *A. fumigatus* Af293 hyphae grown for 16 h in YMM under treatment with 10 μg/mL 5,8-diHODE, 1 μg/mL caspofungin, both treatments, or vehicle only. Examples of lysed germlings are indicated with red arrowheads. Two conidia under treatment with CSPF only are shown at 4-, 8-, 12-, and 16-h post inoculation to visualize the change in light diffraction and cessation of growth used to identify lysed germlings. **b** Percent survival of WT *A. fumigatus* Af293 germlings treated with 1 μg/mL CSPF or 1% DMSO vehicle and 10 μg/mL 5,8-diHODE, 10 μg/mL 8-HODE, or 1% EtOH vehicle after 16 h at 37 °C in YMM. **c** Percent survival of WT *A. fumigatus* Af293 germlings treated with 1 μg/mL CSPF or 1% DMSO vehicle and 5,8-diHODE or 1% EtOH vehicle after 16 h at 37 °C in YMM. **d** Percent survival of WT *A. fumigatus* Af293 germlings treated with 1 μg/mL CSPF or 1% DMSO

vehicle and 10 μg/mL 5,8-diHODE, 8-HODE, 10-HODE, 9-HODE or 1% EtOH vehicle after 16 h growth in YMM. **e** WT A. *fumigatus* CEA10 hyphae grown for 16 h in GMM under treatment with 10 μg/mL 5,8-diHODE, 2 μg/mL caspofungin, both treatments, or vehicle only. Examples of lysed germlings are indicated with red arrowheads. **f** Percent survival of WT *A. fumigatus* CEA10 germlings treated with 2 μg/mL CSPF or 1% DMSO vehicle and 10 μg/mL 5,8-diHODE, 10 μg/mL 8-HODE, or 1% EtOH vehicle after 16 h at 37 °C in GMM. **b, c, d, f** Data points each represent percent survival of 99 germlings assessed in biologically independent samples (*n* = 3) with errors bars representing SEM. **b, f** P values shown were calculated by one-way ANOVA with Tukey's multiple comparisons. **c, d** Conditions with *p* values less than 0.05 calculated by one-way ANOVA with Tukey's multiple comparisons are indicated by distinct letters. Exact *p* values are provided in the source data file.

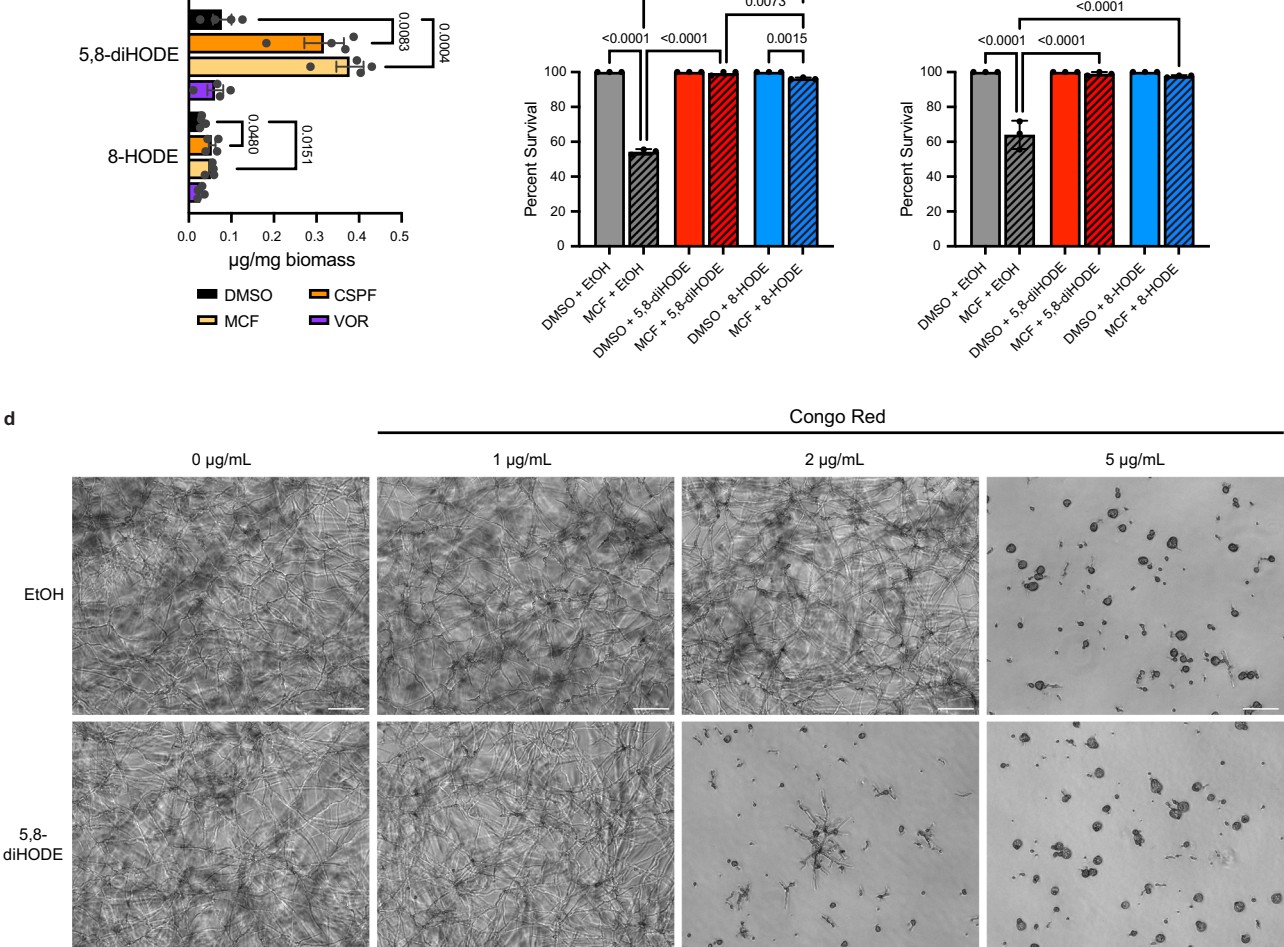

**Fig. 3 | PpoA oxylipin induction and protection are echinocandin specific responses. a** 5,8-diHODE and 8-HODE per mg fungal biomass extracted from mycelial tissue of WT Af293 grown in GMM for 24 h at 37 °C and 250 RPM and an additional 48 h after the addition of 0.02% DMSO, 2 μg/mL caspofungin, 0.5 μg/mg micafungin, or 0.2 μg/mg voriconazole. Data points represent independent culture flasks (*n* = 4) with error bars denoting SEM. *P* values shown were calculated by two-sided Welch's *t*-test. **b** Percent survival of WT *A. fumigatus* Af293 germlings treated with 1 μg/mL MCF or 1% DMSO vehicle and 10 μg/mL 5,8-diHODE, 10 μg/mL 8-HODE, or 1% EtOH vehicle after 16 h at 37 °C in YMM. **c** Percent survival of WT *A. fumigatus*

CEA10 germlings treated with 2 μg/mL MCF or 1% DMSO vehicle and 10 μg/mL 5,8-diHODE, 10 μg/mL 8-HODE, or 1% EtOH vehicle after 16 h at 37 °C in GMM. **b, c** Data points each represent percent survival of 99 germlings assessed in biologically independent samples (*n* = 3) with errors bars representing SEM. *P* values shown were calculated by one-way ANOVA with Tukey's multiple comparisons. **d** Representative brightfield images of WT *A. fumigatus* Af293 treated with increasing concentrations of Congo red plus 10 μg/mL 5,8-diHODE or 1% EtOH vehicle for 16 h at 37 °C in YMM. Scale bar represents 100 microns. Micrographs are representative of three independent experiments.

patients[31]. As in *A. fumigatus*, PpoA produces 5,8-diHODE and 8-HODE from linoleic acid in these species[21,32,33]. Considering the conservation of this oxygenase among aspergilli, we sought to determine whether the induction of protective oxylipin synthesis by PpoA under caspofungin treatment was also conserved.

Northern blot analysis of WT *A. flavus* NRRL3357 and WT *A. nidulans* FGSC A4 revealed that expression of *A.fl.ppoA* and *A.n.ppoA* respectively were induced by caspofungin treatment compared to a DMSO control (Fig. 4a, b). As previously described for *A. fumigatus*, we quantified *A. flavus* WT NRRL 3357 and

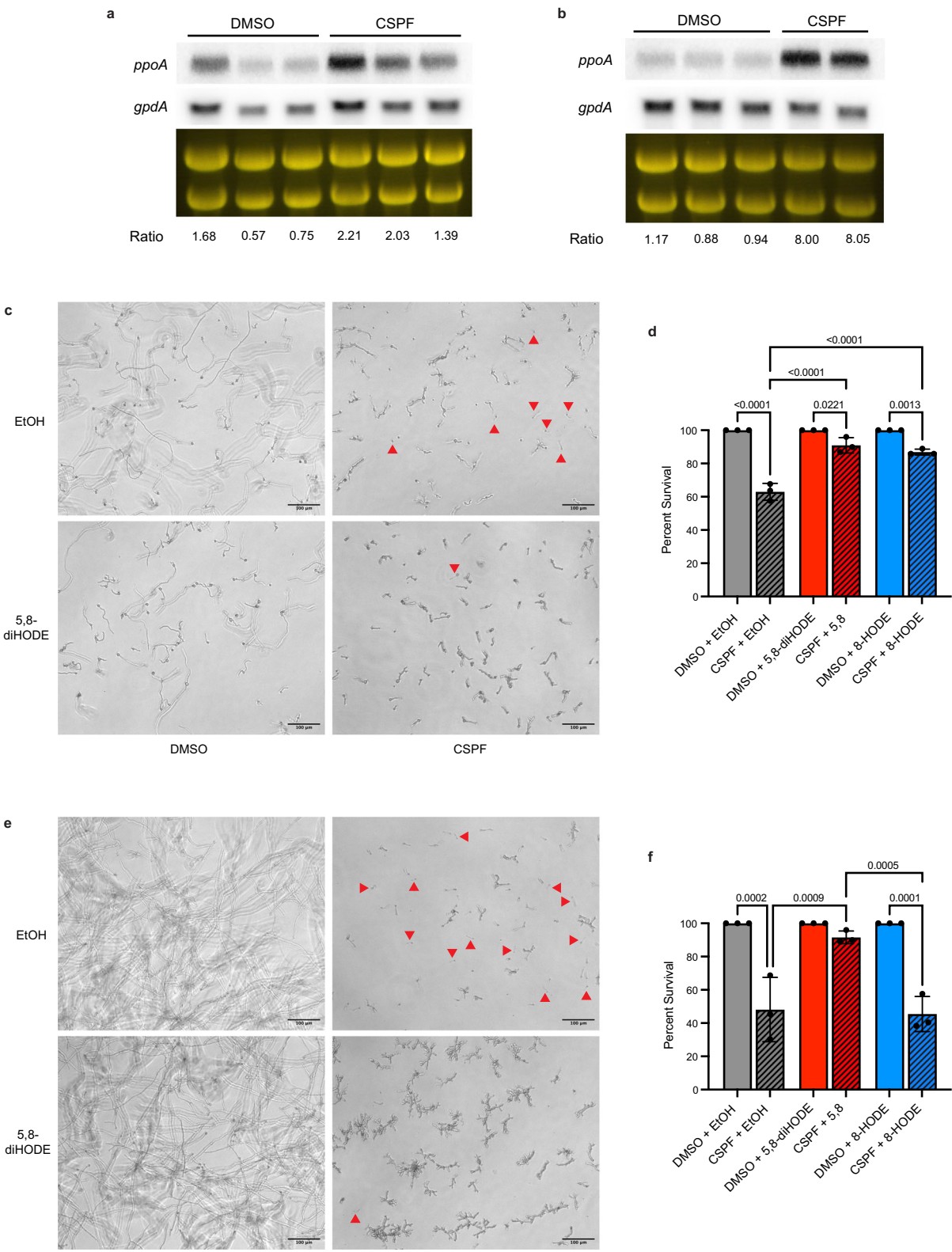

*A. nidulans* WT FGSC A4 germling survival under treatment with caspofungin alone versus 5,8-diHODE/caspofungin cotreatment, or 8-HODE/caspofungin cotreatment with the appropriate vehicle and oxylipin controls. We found that both 5,8-diHODE and 8-HODE improved the survival of *A. flavus* germlings under treatment with 2 μg/mL caspofungin (Fig. 4c, d). 5,8-diHODE also protected *A. nidulans* germlings against caspofungin-mediated tip lysis, but 8-HODE did not improve germling survival in this species at the tested concentration (Fig. 4e, f). Together, these findings suggest that the induction of PpoA oxylipin synthesis is a conserved, protective response among medically relevant aspergilli.

**Fig. 4 | Induction of *ppoA* and protection by 5,8-diHODE under caspofungin treatment is conserved among aspergilli.** Northern blot analysis of (**a**) *A.fl.ppoA* expression in WT *A. flavus* NRRL 3357 and (**b**) *A.n.ppoA* in WT *A. nidulans* FGSC A4 under treatment with 0.025% DMSO vehicle or 1 μg/mL caspofungin. Total RNA was extracted from replicate cultures grown in liquid GMM at 250 RPM for 18.5 h before the addition DMSO or CSPF for 90 min. Loading of 25 μg RNA per sample is shown by ethidium bromide staining of rRNA. Transcript signals of *ppoA* and *gpdA* were measured using ImageJ and signal ratios were normalized by the mean of the DMSO treated samples. **c** WT *A. flavus* NRRL 3357 hyphae grown for 16 h in GMM under treatment with 10 μg/mL 5,8-diHODE, 2 μg/mL caspofungin, both treatments, or vehicle only. Examples of lysed germlings are indicated with red arrowheads.

**d** Percent survival of WT *A. flavus* NRRL 3357 germlings treated with 2 μg/mL CSPF or 1% DMSO vehicle and 10 μg/mL 5,8-diHODE, 10 μg/mL 8-HODE, or 1% EtOH vehicle after 16 h at 37 °C in GMM. **e** WT *A. nidulans* FGSC A4 hyphae grown for 16 h in GMM under treatment with 10 μg/mL 5,8-diHODE, 2 μg/mL caspofungin, both treatments, or vehicle only. Examples of lysed germlings are indicated with red arrowheads. **f** Percent survival of WT *A. nidulans* FGSC A4 germlings treated with 2 μg/mL CSPF or 1% DMSO vehicle and 10 μg/mL 5,8-diHODE, 10 μg/mL 8-HODE, or 1% EtOH vehicle after 16 h at 37 °C in GMM. **d, f** Data points each represent percent survival of 99 germlings assessed in biologically independent samples ($n = 3$) with errors bars representing SEM. *P* values shown were calculated by one-way ANOVA with Tukey's multiple comparisons.

## The oxylipin-responsive transcription factor ZfpA promotes *ppoA* expression and protects against caspofungin-mediated tip lysis

Bearing in mind that the C2H2 transcription factor ZfpA is required for the hyperbranching and chitin responses to 5,8-diHODE[20] and 8-HODE (Supplementary Fig. 7) as well as being involved in tolerance to caspofungin[23], we thought it likely that ZfpA could be required for 5,8-diHODE protection of *A. fumigatus* against the fungicidal activity of echinocandins. We performed germling lysis experiments with WT, Δ*zfpA*, and OE::*zfpA* strains in both Af293 and CEA10 genetic backgrounds. As expected, we found that deletion of ZfpA in both genetic backgrounds rendered germlings hypersusceptible to caspofungin-mediated lysis while overexpression of ZfpA resulted in germlings highly tolerant to caspofungin. Although Δ*zfpA* germling survival in both Af293 and CEA10 backgrounds was significantly improved by 5,8-diHODE cotreatment, this protection did not reach wild type levels (Fig. 5a, b).

Considering that 5,8-diHODE treatment was previously shown to strongly induce *ppoA* expression[20], we thought it possible that ZfpA could regulate expression of this transcript in a manner of positive feedback. Thus, we sought to ask whether ZfpA may be responsible for the upregulation of *ppoA* under treatment with caspofungin. Northern blot analyses revealed that caspofungin induced expression of the *ppoA* transcript in wild-type Af293 but not in a Δ*zfpA* mutant (Fig. 5c). Similarly, treatment with caspofungin failed to induce expression of *ppoA* in a CEA10 Δ*zfpA* mutant in contrast to a robust induction in the WT strain (Fig. 5d). This suggests that ZfpA is required for the induction of *ppoA* by caspofungin presenting an unexpected positive feedback loop whereby ZfpA is required for *ppoA* expression and PpoA oxylipins in turn amplify *zfpA* expression.

We identified the orthologues of *A. fumigatus zfpA* in the *A. flavus* and *A. nidulans* genomes, AFLA_003452 (61.79% aa identity) and AN1500 (64.79% aa identity) respectively. In order to interrogate the functional conservation of these orthologues in each species, we generated deletion and overexpression mutants of both genes (Supplementary Figs. 8 and 9). To assess whether *A. flavus* ZfpA was required for 5,8-diHODE protection against caspofungin-mediated tip lysis, we performed germling lysis experiments with WT, Δ*A.fl.zfpA*, and OE::*A.fl.zfpA* strains. Although Δ*A.fl.zfpA* germlings succumbed to fungicidal tip lysis at a rate similar to WT under caspofungin treatment alone, the addition of 5,8-diHODE cotreatment improved survival in wild type germlings more than in the deletion strain (Fig. 5e). Overexpression of *A.fl.zfpA* drastically improved germling survival under caspofungin treatment which was not increased further by cotreatment with 5,8-diHODE (Fig. 5e). These results suggest that *A.fl.*ZfpA regulates genes involved in tolerance to caspofungin. In *A. nidulans*, deletion of *A.n.zfpA* rendered germlings more susceptible to caspofungin killing by tip lysis while overexpression of *A.n.zfpA* slightly reduced susceptibility compared a wild type control (Fig. 5f). However, germling survival in both mutants was still improved similarly to the wild type strain by the addition of 5,8-diHODE cotreatment (Fig. 5f). These results suggest a role for ZfpA in caspofungin tolerance is conserved among aspergilli, but protection by 5,8-diHODE against fungicidal tip lysis is likely mediated by more than one transcription factor.

Next, we sought to assess whether ZfpA was required for induction of *A.fl.ppoA* and *A.n.ppoA* by caspofungin in *A. flavus* and *A. nidulans*. Northern blot analysis of WT and Δ*A.fl.zfpA* strains revealed that deletion of *A. flavus zfpA* reduced basal *A.fl.ppoA* expression as well as its induction by caspofungin treatment (Fig. 5g). Similarly, the expression of *A.n.ppoA* was reduced in the Δ*A.n.zfpA* strain compared to WT under both DMSO and caspofungin treatment (Fig. 5h). Altogether, these results support a conserved *Aspergillus* 5,8-diHODE/ZfpA relay pathway that confers tolerance to echinocandin fungicidal activity with some variation in the degree of protection between species.

## ZfpA mutant transcriptomes suggest synergistic induction of chitin via MpkA signaling contributes to 5,8-diHODE protection

Considering that ZfpA overexpression phenocopies 5,8-diHODE treatment in morphology and resistance to echinocandin tip lysis, we performed RNA-sequencing of *A. fumigatus* Af293 WT, Δ*zfpA*, and OE::*zfpA* strains in an effort to identify protective effectors downstream of the 5,8-diHODE/ZfpA signaling relay. Relative to wild type, the deletion of *zfpA* resulted in at least two-fold upregulation of 1273 genes compared to downregulation of 348. Conversely, the overexpression of *zfpA* significantly reduced the expression of 1076 genes while it upregulated 648 (Supplementary Data 1). Functional category enrichment analysis of genes up- and downregulated in each mutant suggested potential roles for ZfpA in the regulation of diverse cellular processes including protein homeostasis, cellular transport, and secondary metabolism (Supplementary Fig. 10). Further visual inspection of differentially expressed genes in these mutants revealed dysregulation of 38 genes involved in chitin and related carbohydrate metabolisms (Fig. 6a). Considering that induction of cell wall chitin is a known protective response to caspofungin[18,34], we hypothesized that 5,8-diHODE treatment may synergistically increase cell wall chitin resulting in fortified hyphal tips more resistant to echinocandin tip lysis. To test this, we performed calcofluor white staining of WT *A. fumigatus* Af293 hyphae treated with 1 μg/mL caspofungin or 1% DMSO plus 1% EtOH or 1 μg/mL 5,8-diHODE. Quantification of mean pixel intensity of DAPI-channel hyphal images revealed that cotreatment with caspofungin and 5,8-diHODE significantly increased cell wall chitin above that of either treatment alone (Fig. 6b).

Based on previous studies, we hypothesized that 5,8-diHODE could be activating protective chitin synthesis through the cell wall integrity pathway mediated by MpkA or through calcium/calmodulin signaling via the transcription factor CrzA[35,36]. Thus, we tested both Δ*mpkA* and Δ*crzA* mutants for protection of 5,8-diHODE against caspofungin tip lysis. We found that Δ*mpkA* germlings were hypersusceptible to caspofungin-mediated tip lysis exhibiting less than ten percent survival (Fig. 6c). Cotreatment with 5,8-diHODE improved survival of Δ*mpkA* germlings to about fifteen percent compared to near one-hundred percent survival in the WT and complemented

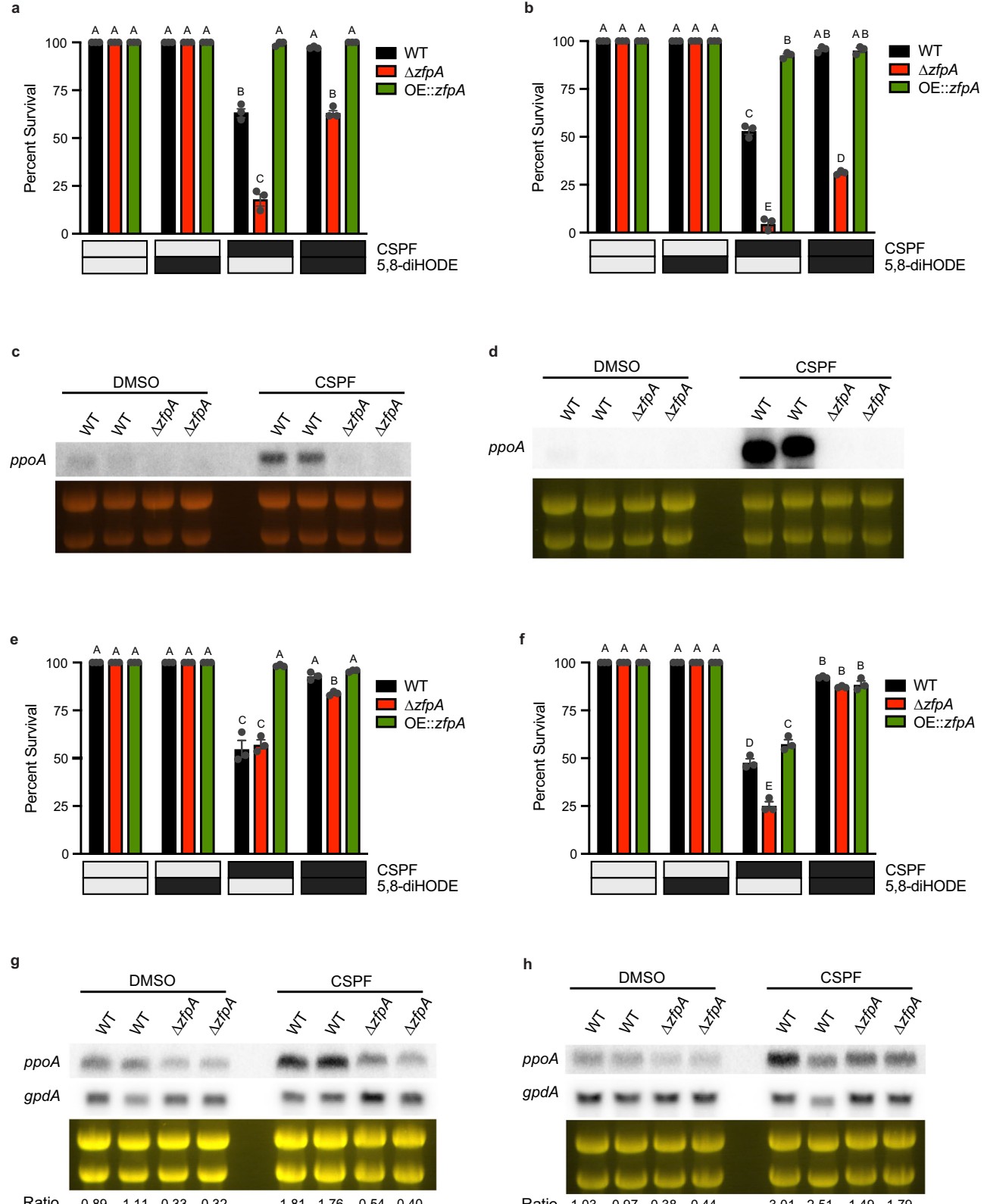

strains (Fig. 6c). This suggests that MpkA is involved in 5,8-diHODE protection against caspofungin mediated tip lysis in addition to other functions of this kinase. Conversely, Δ*crzA* germlings succumbed to caspofungin tip lysis at similar rates to the wild type and complemented strains and were equally protected by the addition of 5,8-diHODE cotreatment (Fig. 6d). Taken together, these results suggest that synergistic activation of chitin synthetic processes via MpkA

signaling may contribute to 5,8-diHODE protection against echinocandin tip lysis in *A. fumigatus*.

## Discussion

Echinocandins are lipopeptide natural products of a subset of fungi belonging to either the soil inhabiting Eurotiomycetes (*Aspergillus* and other genera) or Leotiomycetes[10,11,37]. The finding of their efficacy in

**Fig. 5 | ZfpA is a mediator of 5,8-diHODE protection against echinocandins in aspergilli via a positive feedback loop with PpoA. a** Percent survival of *A. fumigatus* Af293 WT, Δ*zfpA*, and OE::*zfpA* germlings treated with 1 μg/mL CSPF or 1% DMSO vehicle and 10 μg/mL 5,8-diHODE or 1% EtOH vehicle after 16 h at 37 °C in YMM. **b** Percent survival of *A. fumigatus* CEA10 WT, Δ*zfpA*, and OE::*zfpA* germlings treated with 2 μg/mL CSPF or 1% DMSO vehicle and 10 μg/mL 5,8-diHODE or 1% EtOH vehicle after 16 h at 37 °C in GMM. Northern blot analysis of *ppoA* expression in (**c**) Af293 WT and Δ*zfpA* strains or (**d**) CEA10 WT and Δ*zfpA* strains under treatment with 0.025% DMSO vehicle or 1 μg/mL caspofungin. Total RNA was extracted from replicate cultures grown in liquid GMM at 250 RPM for 18.5 h before the addition of DMSO or CSPF for 90 min. Loading of 25 μg RNA per sample is shown by ethidium bromide staining of rRNA. **e** Percent survival of *A. flavus* WT, Δ*A.fl.zfpA*, and OE::*A.fl.zfpA* germlings treated with 2 μg/mL CSPF or 1% DMSO vehicle and 10 μg/mL 5,8-diHODE or 1% EtOH vehicle after 16 h at 37 °C in GMM. **f** Percent survival of *A. nidulans* WT, Δ*A.n.zfpA*, and OE::*A.n.zfpA* germlings treated with 2 μg/mL CSPF or 1% DMSO vehicle and 10 μg/mL 5,8-diHODE or 1% EtOH vehicle after 16 h at 37 °C in GMM. Northern blot analysis of (**g**) *A.fl.ppoA* expression in WT and Δ*A.fl.zfpA A. flavus* or (**h**) *A.n.ppoA* in WT and Δ*A.n.zfpA A. nidulans* under treatment with 0.025% DMSO vehicle or 1 μg/mL caspofungin. Total RNA was extracted from replicate cultures grown in liquid GMM at 250 RPM for 18.5 h before the addition of DMSO or CSPF for 90 min. Loading of 25 μg RNA per sample is shown by ethidium bromide staining of rRNA. Transcript signals of *ppoA* and *gpdA* were measured using ImageJ and signal ratios were normalized by the mean of the WT DMSO treated samples. **a**, **b**, **e**, **f** Data points each represent percent survival of 99 germlings assessed in biologically independent samples (*n* = 3) with errors bars representing SEM. Conditions with *p* values less than 0.05 calculated by two-way ANOVA with Tukey's multiple comparisons are indicated by distinct letters. Exact *p* values are provided in the source data file.

controlling the growth of some human pathogenic fungi led to the development of semisynthetic therapeutic derivatives with caspofungin and micafungin most frequently used[12]. An interesting feature of antifungal metabolites produced by fungi is the presence of self-protective mechanisms in the producing fungus absent in competing fungi[38]. While lacking these specific self-protection mechanisms, non-producing fungi can evolve other means to tolerate antifungals, particularly those encountered in their ecological niche. The fungistatic rather than fungicidal effect of echinocandins against *A. fumigatus*[14]—limiting their clinical use against IA—suggests that just such a tolerance program is operable in this species. Here, we present evidence supporting a conserved role for the PpoA oxylipins 5,8-diHODE and 8-HODE in activating an echinocandin tolerant growth program characterized by increased branching, septation, and cell wall chitin deposition. This program is present in all *A. fumigatus* strains tested in this study and extends to two other pathogenic *Aspergillus* spp.

Following from our recent study which found the fungal oxylipin 5,8-diHODE to induce hyphal growth morphologically similar to growth under caspofungin treatment[20], we investigated a role for this oxylipin as an endogenous signal activating the *A. fumigatus* echinocandin response. We recapitulated published findings that treatment with 5,8-diHODE and caspofungin both increase hyphal branching, septation, and cell wall chitin[15,17–20]. Further, we found that treatment with 8-HODE, the precursor oxylipin to 5,8-diHODE, also induced these phenotypic responses in developing hyphae. Our first key discovery in this study shows that caspofungin robustly induced the expression of the *ppoA* gene and subsequent production of 5,8-diHODE in WT *A. fumigatus* mycelia. An increase in the precursor oxylipin 8-HODE was also detectable in extracts of the caspofungin treated tissue to a lesser degree. Additionally, an induction of 5,8-diHODE, but not 8-HODE, was detected in extracts from culture supernatants. The smaller amounts of 8-HODE compared to 5,8-diHODE are likely due to rapid enzymatic conversion by PpoA to the endpoint metabolite.

In vitro, two related but distinct inhibitory effects on hyphal growth of *A. fumigatus* can be observed under caspofungin treatment. Beginning even at low concentrations, the inhibition of beta-1,3-glucan synthase disrupts normal cell wall synthesis at growing tips causing the apical compartments of hyphae to lyse. Although echinocandins are considered to be fungistatic against aspergilli, this lysis of the apical tips is known to be a fungicidal effect of these drugs[14,15,17]. The fungicidal capacity of echinocandins beyond the apical compartment is blocked by the plugging of hyphal septa[16,39]. Following hyphal tip lysis, a new growing tip can emerge from the apical most septum resulting in what are referred to as intrahyphal hyphae[14,15,40]. In the absence of septa, caspofungin exerts fungicidal activity on the entire hyphal compartment[16]. At higher concentrations, tip lysis continues to occur but hyphal growth is also fungistatically inhibited resulting in stubby, highly branched, highly septate, chitin-rich microcolonies[15,17,18]. At effective concentrations, we and others observed that early tip lysis of one or both germ tubes extending from the swollen spore body resulted

in death of a significant proportion of germlings[19]. It is likely that caspofungin exerts fungicidal activity on these germlings similar to its activity against aseptate hyphae by lysing the germ tube tips before the formation of the first septum or by lysing all cellular compartments of early hyphae with few septa. We found that up to sixty percent of wild-type Af293 germlings under treatment with an effective concentration of 1 μg/mL caspofungin succumbed to the fungicidal activity of this drug, but when conidia were cotreated with 10 μg/mL 5,8-diHODE, the survival rate was restored back to nearly one hundred percent. A similar but slightly weaker protective effect was also observed under cotreatment with 8-HODE. It is known that distinct isolates of *A. fumigatus* exhibit disparate responses and susceptibilities to caspofungin[24,28,29]. Therefore, our findings that 5,8-diHODE conferred protection against caspofungin-mediated lysis in all three common *A. fumigatus* laboratory isolates Af293, CEA10, and AfS35, in addition to the caspofungin tolerant clinical isolate CM7555, support a conserved role for this oxylipin in the *A. fumigatus* caspofungin response.

While caspofungin is perhaps the best characterized echinocandin in terms of *A. fumigatus* biology, the drugs micafungin and anidulafungin also inhibit beta-1,3-glucan synthase[12,41]. As with caspofungin, treatment with micafungin induced the production of 5,8-diHODE detectable in extracts from WT *A. fumigatus* Af293 mycelial tissue. Conversely, treatment with the non-echinocandin antifungal, voriconazole, did not elicit any detectable increase in 5,8-diHODE. Furthermore, we found that 5,8-diHODE and 8-HODE also conferred protection to WT Af293 and CEA10 germlings against micafungin-mediated lysis. However, 5,8-diHODE cotreatment was not protective against inhibition by the cell wall perturbing dye Congo red. This is likely due to mechanistic differences in the inhibition by Congo red compared to echinocandin drugs[42]. Together, these data indicate that the induction of PpoA oxylipin synthesis is a protective response specific to echinocandin antifungal assault.

Other disease-causing aspergilli are also susceptible to inhibition by echinocandin antifungals[43], including *A. flavus*, another causative agent of IA[44], and the model organism *A. nidulans* which can cause severe disease in chronic granulomatous disease patients[31]. Orthologues of the *A. fumigatus* PpoA protein are also found in these *Aspergillus* species and function to convert linoleic acid into 8-HODE and 5,8-diHODE[33]. Notably, 5,8-diHODE was found to trigger an increase in lateral branching in *A. flavus* hyphae similar to that observed in *A. fumigatus*[20]. Here, we found that caspofungin treatment also induced expression of *A.fl.ppoA* and *A.n.ppoA* in *A. flavus* NRRL 3357 and *A. nidulans* FGSC A4 respectively. 5,8-diHODE cotreatment also protected against tip lysis in WT *A. flavus* and WT *A. nidulans* to significantly improve germling survival under caspofungin treatment. Together, these findings suggest that a conserved role for PpoA oxylipins in the tolerance of echinocandin fungicidal activity extends across the genus *Aspergillus*.

In *A. fumigatus*, the C2H2 zinc finger transcription factor ZfpA is known to be an oxylipin-responsive regulator of branching, septation,

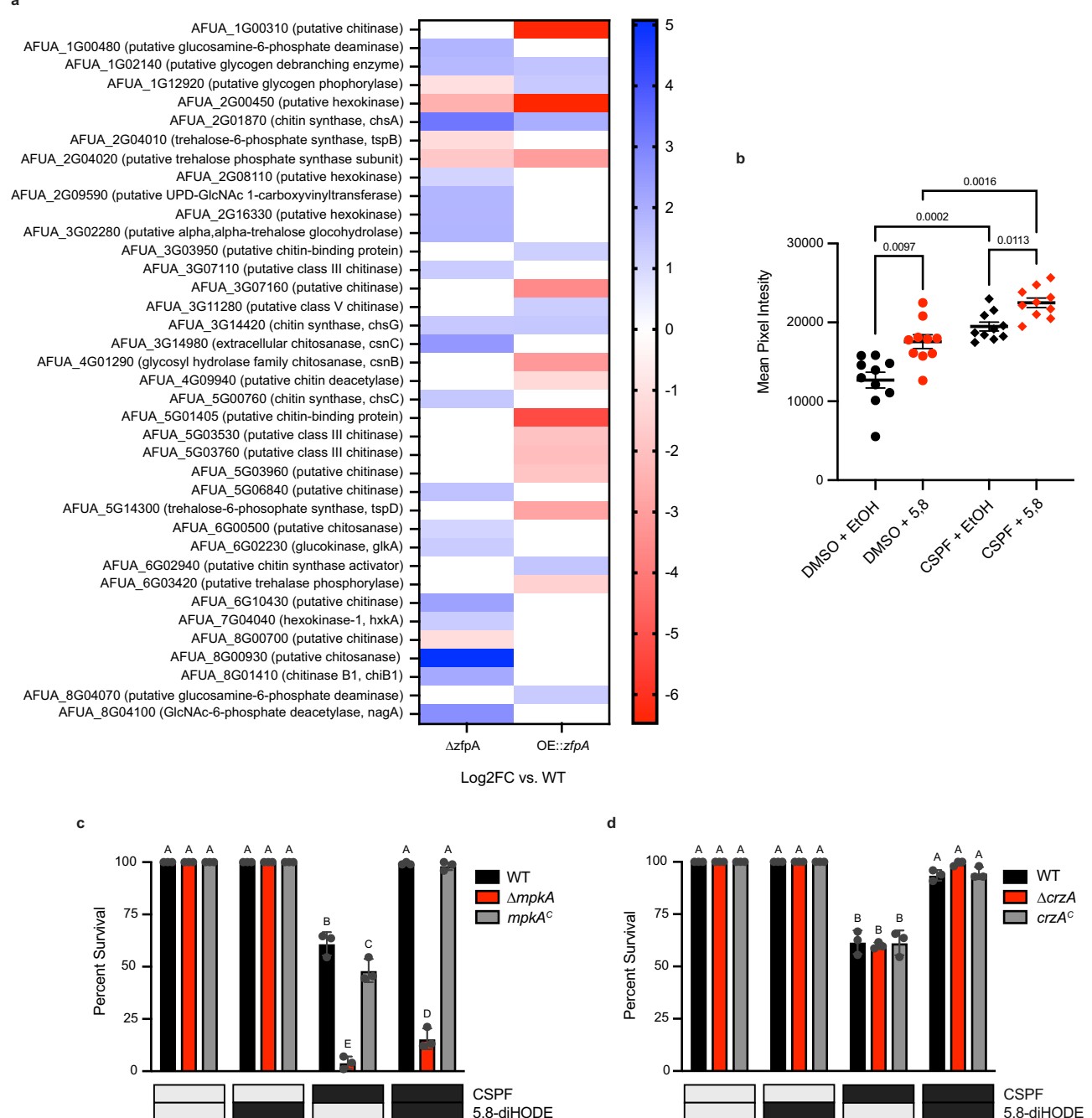

**Fig. 6 | Synergistic increase in cell wall chitin via MpkA signaling may contribute to 5,8-diHODE protection against caspofungin tip lysis. a** Log2 fold change of differentially expressed chitin and related carbohydrate metabolic genes in ΔzfpA and OE::zfpA mutants relative to WT Af293. Differentially expressed genes were defined as |Log2FC| ≥ 1 and p. adj. < 0.01. **b** Mean calcofluor white intensity per pixel of WT Af293 hyphae grown for 15 h in GMM with 1% DMSO or 1 μg/mL caspofungin and 1 μg/mL 5,8-diHODE or EtOH for 15 h before staining and epi-fluorescent imaging. Data points represent individual hyphae (n = 10) with error bars representing SEM. P values shown were determined by Browne-Forsythe and Welch ANOVA with Dunnett's T3 multiple comparisons tests. **c** Percent survival of

A. fumigatus CEA10 WT, ΔmpkA, and mpkA^C germlings treated with 2 μg/mL CSPF or 1% DMSO vehicle and 10 μg/mL 5,8-diHODE or 1% EtOH vehicle after 16 h at 37 °C in GMM. **d** Percent survival of A. fumigatus CEA10 WT, ΔcrzA, and crzA^C germlings treated with 2 μg/mL CSPF or 1% DMSO vehicle and 10 μg/mL 5,8-diHODE or 1% EtOH vehicle after 16 h at 37 °C in GMM. **c**, **d** Data points each represent percent survival of 99 germlings assessed in biologically independent samples (n = 3) with errors bars representing SEM. Conditions with p values less than 0.05 calculated by two-way ANOVA with Tukey's multiple comparisons are indicated by distinct letters. Exact p values are provided in the source data file.

and chitin deposition[20]. It is also known that expression of *zfpA* is upregulated under caspofungin treatment where it plays a protective role in both the Af293 and CEA10 genetic backgrounds[22–24]. We found that deletion of ZfpA in both Af293 and CEA10 backgrounds produced germlings hypersensitive to caspofungin-mediated lysis while OE::*zfpA*

germlings in both backgrounds phenocopied treatment with 5,8-diHODE exhibiting resistance to caspofungin tip lysis with nearly one-hundred percent survival of germlings under caspofungin treatment alone. These findings provide evidence for ZfpA as a key regulator of genes involved in 5,8-diHODE mediated echinocandin tolerance.

Nonetheless, protection of both Af293 and CEA10 Δ*zfpA* germlings to wild-type levels could not be achieved with 5,8-diHODE cotreatment, suggesting that while ZfpA is required to obtain the full protective effect of 5,8-diHODE against echinocandin damage, other response factors are likely involved. ZfpA was found to be the key transcription factor in mediating developmental responses to 5,8-diHODE, but several other transcription factors were also transcriptionally induced by treatment with this oxylipin[20].

Interestingly, *ppoA* was also found to be one of the most highly upregulated genes under exogenous treatment with 5,8-diHODE suggesting an auto-inductive effect possibly via ZfpA[20]. Given this, we posited that ZfpA may be responsible for the initial PpoA induction under caspofungin treatment. Indeed, we found that deletion of *zfpA* in both Af293 and CEA10 genetic backgrounds abolished the caspofungin-responsive increase in *ppoA* expression observed in wild-type strains. These findings suggest that ZfpA is required to activate *ppoA* transcription under caspofungin treatment resulting in increased production of 5,8-diHODE and 8-HODE which by positive feedback further activate ZfpA and other yet unidentified transcription factors to direct an echinocandin tolerant growth program.

*A. flavus* and *A. nidulans* also possess orthologues of the *A. fumigatus* ZfpA transcription factor. We generated deletion and overexpression mutants in order to assess whether ZfpA performs conserved functions in these species. While equally susceptible to tip lysis by caspofungin alone, Δ*A.fl.zfpA* demonstrated a slight diminution of 5,8-diHODE protection compared to the wild type. Furthermore, the OE::*A.fl.zfpA* mutant was highly resistant to tip lysis phenocopying 5,8-diHODE treatment as observed previously for *A. fumigatus*. Similarly, *A. nidulans* Δ*A.n.zfpA* germlings were more susceptible to caspofungin tip lysis and OE::*A.n.zfpA* germlings slightly more resistant than wild type. However, all three strains were comparably protected by 5,8-diHODE cotreatment suggesting other transcription factors may also contribute to oxylipin protection in this species. Additionally, we found that *A.fl.zfpA* and *A.n.zfpA* were required for full expression of *ppoA* orthologues in these species including in response to caspofungin. These results present robust data that a protective ZfpA/PpoA signaling relay plays a conserved role in the echinocandin response of pathogenic aspergilli.

Considering that ZfpA overexpression phenocopies 5,8-diHODE treatment in morphology and resistance to echinocandin tip lysis, we performed RNA-sequencing of *A. fumigatus* Af293 WT, Δ*zfpA*, and OE::*zfpA* strains in an effort to identify protective effectors downstream of the 5,8-diHODE/ZfpA signaling relay. Analysis of DEGs in these mutants revealed altered expression of numerous genes involved in chitin metabolism suggesting that increases in cell wall chitin via ZfpA underpins the protective effect of PpoA oxylipins against echinocandin tip lysis. In agreement with this hypothesis, we found that cotreatment of WT *A. fumigatus* Af293 hyphae with both caspofungin and 5,8-diHODE significantly increased chitin levels compared to either treatment alone. These findings support a model whereby caspofungin and 5,8-diHODE synergistically activate distinct chitin metabolic genes resulting in a fortified cell wall with hyphal tips resistant to echinocandin-mediated lysis (Fig. 7).

The cell wall integrity and calcium-calmodulin signaling pathways are known to be the two main regulators of chitin biosynthesis in aspergilli[45]. Therefore, in an effort to gain mechanistic insight into the chitin response to 5,8-diHODE, we assessed caspofungin germling lysis in Δ*mpkA* and Δ*crzA* mutants of CEA10. Echinocandins activate the cell wall integrity pathway which culminates in the activation of the mitogen activated protein kinase MpkA[46]. Given that MpkA plays a crucial role in the response to cell wall stress[35,47], we posited that 5,8-diHODE may protect against tip lysis via synergistic activation of this MAPK cascade. Indeed, the Δ*mpkA* mutant was extremely susceptible to germling killing by caspofungin tip lysis even with 5,8-diHODE cotreatment compared to wild type and complemented strains.

However, survival was still significantly improved by 5,8-diHODE suggesting that MpkA is important in the protective response to 5,8-diHODE but not the sole contributor. It is worth investigating how 5,8-diHODE might alter the phosphorylation states of the kinases involved in the CWI pathway. Under high levels of caspofungin, calcium-calmodulin signaling activates the calcium responsive transcription factor CrzA via dephosphorylation by calcineurin[34,36,48,49]. Dephosphorylated CrzA translocates to the nucleus to induce the expression of chitin synthetic genes important for echinocandin tolerance[36]. However, both tip lysis by caspofungin alone and protection under 5,8-diHODE cotreatment in Δ*crzA* germlings were indistinguishable from wild type and complemented strains. This suggests that, in CEA10, the protective response to 5,8-diHODE does not require the transcription factor CrzA. However, it is known that the role of CrzA in distinct isolates of *A. fumigatus* is highly divergent where other transcription factors may play compensatory roles[24,49]. Interestingly, one study comparing the role of CrzA in Af293 and CEA10 showed that induction of *zfpA* expression by caspofungin can be abolished by deletion of *crzA* in both backgrounds[24], suggesting that ZfpA may be differently activated by caspofungin and 5,8-diHODE.

In summary, this work provides clear evidence that a conserved, endogenous mechanism protects *Aspergillus* spp. from echinocandin damage. Our model presents a ZfpA/PpoA signaling relay that protects growing tips by reinforcing chitin deposition in hyphal cell walls (Fig. 7). We speculate that this protective oxylipin signaling mechanism may extend to other filamentous fungi. PpoA-like proteins have been identified in all assessed filamentous ascomycetes[33,50] and a previous study on ZfpA showed this transcription factor to be broadly conserved as well[51]. Furthermore, 7,8-diHODE synthesized by the *Magnaporthe oryzae* PpoA-like enzyme also induced lateral branching in *A. fumigatus* similarly to 5,8-diHODE[20]. It is plausible that an oxylipin-directed growth program evolved in filamentous fungi as a protective strategy against echinocandin producing competitors in shared ecological niches.

## Methods

### Fungal strains, media, and culture conditions

Wild type *A. fumigatus* Af293, CEA10, and AfS35 are common clinical laboratory strains available through ATCC. *A. flavus* NRRL 3357 and *A. nidulans* FGSC A4 are also ATCC available. All fungal strains used in this study are described in Supplementary Table 1. Strains were maintained in 25-50% glycerol stock suspensions at −80 °C and grown on glucose minimal media (GMM) plates at 37 °C for 3 days with the exception of *A. flavus* grown for 4 days at 30 °C. Conidia were collected into sterile water with 0.01% Tween 80. GMM plates and broth were prepared as previously described[52]. YMM broth was prepared by supplementing GMM with 0.1% w/v yeast extract. YMM was utilized for all experiments with *A. fumigatus* Af293 strains in static liquid culture due to an observation of less than one hundred percent germination in GMM.

### Construction of *A. flavus* and *A. nidulans* zfpA mutants

To generate *A.fl.zfpA* and *A.n.zfpA* deletion strains of *A. flavus* or *A. nidulans*, two 1 kb DNA fragments immediately upstream and downstream of each *zfpA* open reading frames (ORF), were amplified by PCR from *A. flavus* NRRL3357 or *A. nidulans* RDIT9.32 genomic DNA, and were fused to a 2 kb *A. parasiticus pyrG* fragment from pJW24[53] using double joint PCR[54]. Fungal transformation of *A. flavus* TJES19.1 or *A. nidulans* TJMP1.49 was performed following previously described approach[55]. Transformants were confirmed for targeted replacement of the native locus by PCR (data not shown) and Southern blotting using *Eco*RI digest for *A. flavus* or *Pvu*II digest for *A. nidulans* with dCTP-αP[32] labeled 5′ and 3′ flanks of each knockout construct to get TJW320.12 from TJES19.1 (Supplementary Fig. 8) and TJW326.4 from RJMP1.49 (Supplementary Fig. 9). TJW326.4 was crossed with RTMH207.13 sexually to create RJW343.2.

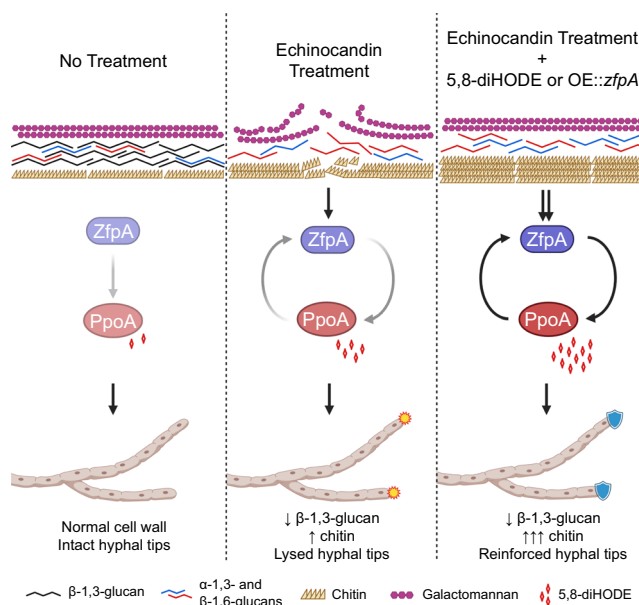

**Fig. 7 | Model of 5,8-diHODE mediated protection against echinocandin damage in *A. fumigatus*.** In the absence of cell wall stress, ZfpA mediates basal *ppoA* expression and normal hyphal growth occurs. Treatment with echinocandins drastically alters cell wall organization by direct inhibition of beta−1,3-glucan synthesis and a compensatory increase in chitin synthesis ultimately resulting in fungicidal lysis of hyphal tips. This disruption of normal cell wall architecture activates a ZfpA/PpoA signaling relay protective against hyphal tip lysis. Cotreatment with 5,8-diHODE or *zfpA* overexpression during echinocandin exposure amplifies activation of the ZfpA/PpoA signaling relay which results in synergistic cell wall chitin induction in a manner that stabilizes hyphal tips and prevents echinocandin mediated tip lysis. Schematic was created with BioRender.com.

To generate *A.fl.zfpA* and *A.n.zfpA* overexpression strains, two 1 kb fragments immediately upstream and downstream of *zfpA* translational start site were amplified by PCR from *A. flavus* NRRL3357 or *A. nidulans* RDIT9.32 genomic DNA. *A. parasiticus pyrG::A. nidulans gpdA(p)* as the selectable marker and overexpression promoter was amplified from the plasmid pJMP9[56]. These three fragments were fused by double joint PCR[54] and transformed into *A. flavus* TJES19.1 or *A. nidulans* TJMP1.49 to create strain TJW321.4 or TJW327.7 respectively. Single integration of the transformation construct was confirmed by PCR (data not shown) and Southern blotting using *Nco*I digest for *A. flavus* or *Pvu*II digest for *A. nidulans* with both dCTP-αP[32] labeled 5′ and 3′ flanks of each over expression construct (Supplementary Figs. 8 and 9). TJW327.7 was crossed with RTMH207.13 sexually to create RJW344.1. All of primers for this study is listed in Supplementary Table 2.

### Antifungal drugs and oxylipins
All antifungals including caspofungin, micafungin, and voriconazole were purchased from ApexBio and stored as 10 mg/mL stock concentrations in DMSO at −20 °C. Purified fungal oxylipins 5(*S*),8(*R*)-dihydroxydecadienoic acid (5,8-diHODE) and 8(*R*)-hydroxydecadienoic (8-HODE) were prepared as previously described[57]. Oxylipin stocks were stored at −20 °C in EtOH.

### Calcofluor white staining for hyphal imaging
Approximately 50 *A. fumigatus* WT Af293 conidia were inoculated into three wells of a cover glass bottom 96-well plate in 0.1 mL GMM with the specified treatment or vehicle control. The plates were incubated for 20 h at 37 °C before staining with a 1:1 ratio of calcofluor white stain (Sigma-Aldrich) and 10% KOH per manufacturers protocol. Stained hyphae were rinsed twice in sterile PBS before DAPI channel

fluorescent imaging on a Nikon Eclipse Ti Inverted Microscope using a Nikon Plan Fluor 20X Ph1 DLL objective.

### Quantification of lateral branching
In total, 1000 conidia in 100 μL GMM with the designated treatment were inoculated into triplicate wells of a 96-well plate. The plate was incubated on a Nikon Eclipse Ti Inverted Microscope in a heated microscope enclosure (OKO Labs, Burlingame, CA) at 37 °C for 15 h before imaging. Microscope frames were set on four germlings per well and images were acquired every 15 min for 6 h using a Nikon Plan Fluor 20X Ph1 DLL objective out to 22 h post inoculation. Lateral branching of eight hyphae per condition at 20 h post inoculation was quantified manually using the NIS-Elements AR Software package (Version 5.30) to measure the distance between each lateral branch and dividing the total distance from the first to last branch by the number of branches.

### CFW cell wall chitin quantification
In total, 500 conidia of WT *A. fumigatus* Af293 in 100 μL GMM with 1% EtOH, 1 μg/mL 5,8-diHODE, 1 μg/mL 8-HODE, 1% DMSO, or 1 μg/mL caspofungin were inoculated into wells of a coverslip glass bottom 96-well plate. After incubation at 37 °C for 15 h, the hyphae were rinsed once with PBS before staining for 5 min with 0.1 mg/mL calcofluor white (CFW) in PBS. Stained hyphae were washed twice for 5 min with fresh PBS before DAPI channel fluorescent imaging on a Nikon Ti2E inverted microscope with a Nikon CFI Plan Apochromat Lamda D 20X objective. Mean CFW signal intensity per pixel of ten hyphae per condition was measured using FIJI (v2.14.0/1.54f) by using the "Huang" auto-thresholding and wand tracing tool functions to select the hyphal area before quantifying mean pixel intensity.

### RNA extraction and Northern blotting
Conidia were inoculated at 10⁶ spores/mL into 50 mL GMM and incubated at 37 °C and 250 RPM for 18.5 h before adding caspofungin (1 μg/mL) or DMSO (0.025%) and incubating for 90 min more. Tissue was collected into sterile miracloth, flash frozen, and lyophilized. Total RNA was extracted using QIAzol lysis reagent (Qiagen) per manufacturer's protocol with the addition of a phenol:chloroform:isoamyl alcohol (25:24:1) extraction step before RNA precipitation. RNA purity and concentration were assessed by nanodrop. Approximately 25 μg of each sample were run in a 1.3% agarose 1.5% formaldehyde gel and transferred to an Amersham Hybond N+ Membrane. Membranes were hybridized with a doubly labeled dATP-αP[32] dCTP-αP[32] probe complementary to a 500 bp to 1 kb region of the gene of interest lacking any predicted introns.

### Fungal oxylipin extraction
*A. fumigatus* WT Af293 conidia were inoculated at 5 × 10⁶ spores per mL into 50 mL GMM and incubated at 37 °C and 250 RPM for 24 h before the addition of caspofungin (2 μg/mL), micafungin (0.5 μg/mL), voriconazole (0.2 μg/mL), or DMSO (0.0002%, v/v) and an additional 48-h incubation. Supernatants were filtered into 250 mL glass bottles by collecting mycelial tissue into sterile miracloth. The collected tissue was press dried, flash frozen, and lyophilized. Supernatants were extracted overnight for oxylipins as previously described with 100 mL ethyl acetate (EA):methanol (MeOH):dichloromethane (DCM) (8:1:1) mixed organic solvent[20]. The organic phase was collected using a separatory funnel and evaporated to dryness using a Buchi Rotovap R-210. Lyophilized tissue was weighed to determine dry biomass before being homogenized in 10 mL sterile water with 0.01% formic acid. Homogenized tissue was extracted overnight in 120 mL mixed organic solvent EA:MeOH:DCM. Tissue and debris were filtered out using Whatman paper filters, and equal volume milli-Q water was added to the mixed organic solvent in a separatory funnel. The organic layer was collected after shaking and venting twice and evaporated to

dryness. Total extracts were dissolved in 4 mL MeOH and transferred to pre-weighed 20 mL glass scintillation vials and evaporated to dryness again. Dried extracts were weighed and stored at −80 °C. Extracts were dissolved at 1 mg/mL in MeOH for analysis by UHPLC–HRMS/MS.

## UHPLC–HRMS/MS analysis

Ultra high pressure liquid chromatography–high resolution mass spectrometry (UHPLC–HRMS) data were acquired using a Thermo Scientific Q Exactive Orbitrap mass spectrometer coupled to a Vanquish UHPLC operated in both positive and negative ionization modes. All solvents used were of spectroscopic grade. Each sample was filtered with 0.2 μm syringe filter. A Waters XBridge BEH-C18 column (2.1 × 100 mm, 1.7 μm) was used with acetonitrile (0.1% formic acid) and water (0.1% formic acid) as solvents at a flow rate of 0.2 mL/min. The screening gradient method for the samples is as follow: Starting at 55% organic hold for 1 min, followed by a linear increase to 98% organic over 18 min, holding at 98% organic for 2 min, for a total of 21 min. A quantity of 10 μL of each sample was injected into the system for the analysis. Purified 5,8-diHODE and 8-HODE were used as standards. For the quantification, standard curves for 5,8-diHODE and 8-HODE were calculated based on intensities from 6 different concentrations of each purified oxylipin (5, 2.5, 1.25, 0.625, 0.3125, and 0.15625 ppm).

## Time-lapse imaging of *A. fumigatus* hyphae

Approximately 1000 *A. fumigatus* conidia in liquid YMM or GMM with caspofungin, 5,8-diHODE, both compounds, or vehicle only were inoculated into three wells each of a 96-well plate. The plate was incubated on a Nikon Eclipse Ti Inverted Microscope in a heated microscope enclosure (OKO Labs, Burlingame, CA) at 37 °C for 3 h before imaging. Images were acquired of approximately twenty-five spores in each well every 15 min for 21 h using a Nikon Plan Fluor 10X Ph1 DLL objective. Images from one well of each condition were converted into a video file to allow visualization of developing hyphae out to 24 h.

## Germling lysis experiments

For each experimental condition, ~4000 conidia were inoculated into three independent wells of a 96-well plate in 0.1 mL liquid YMM or GMM. Treatments included CSPF, MCF, or DMSO vehicle plus 5,8-diHODE, 8-HODE, 10-HODE, 9-HODE or EtOH vehicle. The plate was incubated on a Nikon Eclipse Ti Inverted Microscope in a heated microscope enclosure (OKO Labs, Burlingame, CA) at 37 °C for 3 h before imaging. During this incubation, three XY imaging positions were set in each well with at least 33 visible spores in each frame. Images were acquired every 15 min at each XY position for 21 h using a Nikon Plan Fluor 10X Ph1 DLL objective. Ninety-nine spores from the three frames in each well were selected blindly at time zero (3 h post inoculation) and annotated in NIS-Elements AR software package (Version 5.30) to be assessed for lysis out to 16 h post inoculation. The number of annotated germlings lysed by 16 h post inoculation was manually counted for each of the three wells per condition. Complete lysis of germlings was determined by the loss of normal diffraction of light compared to intact hyphae and confirmed by complete cessation of growth out to 24 h.

## Congo red imaging

For each experimental condition, ~4000 WT *A. fumigatus* Af293 conidia were inoculated into three independent wells of a 96-well plate in 0.1 mL liquid YMM. Treatments included concentrations of Congo Red of 0, 1, 2, and 5 μg/mL Congo red with wither 10 μg/mL 5,8-diHODE or 1% EtOH. The plate was at 37 °C for 16 h before washing twice with PBS and imaging. Representative images of hyphal growth were acquired in each well using a Nikon Plan Fluor 10X Ph1 DLL objective.

## RNA sequencing and differential gene expression analysis

*A. fumigatus* Af293 wild type, Δ*zfpA*, and OE::*zfpA* conidia were inoculated at $10^6$ spores per mL in liquid glucose minimal media and incubated at 37 °C shaking at 250 RPM overnight. For each sample, the total tissue was combined from two 50 mL cultures flash frozen and lyophilized. Total RNA was extracted using QIAzol Lysis Reagent (Qiagen) according to the manufacturer's instructions with an additional phenol:chloroform:isoamyl alcohol (24:1:1) extraction step before RNA precipitation. Total RNA extraction was completed per manufacturer's instructions with an additional phenol:chloroform:isoamyl alcohol (24:1:1) extraction step before RNA precipitation. Total RNA samples were further cleaned up using RNeasy Mini Kit with on-column DNase digestion (Qiagen) per manufacturer's protocol. RNA integrity was tested via nanodrop, gel electrophoresis, and the Agilent 2100 Bioanalyzer. Library preparation and RNA sequencing were performed by Novogene, Inc. using the TruSeq Stranded mRNA Library Prep Kit and Illumina Novaseq 6000 Platform. Adapter and low- quality reads were removed, and clean reads were mapped to the annotated genome of *A. fumigatus* Af293 obtained from FungiDB (release 52) using HiSAT2 (v2.0.5). The read-count table was processed through DESeq2 (v1.20.0) in R to identify differentially regulated genes between mutant and wild type samples. Significantly differentially expressed genes (DEGs) in Δ*zfpA* and OE::*zfpA* mutants were defined as having |Log2FC| ≥ 1 and a Benjamini-Hochberg adjusted $p < 0.01$. Functional catalog (FunCat[58]) enrichment analysis was performed on significantly up- and downregulated genes in each mutant using FungiFun2[59] with a Benjamini-Hochberg false detection rate of 0.05.

## Caspofungin radial growth testing

To assess sensitivity to caspofungin, square GMM plates supplemented with DMSO, 0.125, 0.25, 0.5, 1, or 8 μg/mL caspofungin with were inoculated with $10^5$, $10^4$, $10^3$, and $10^2$ spores in a volume of 2 μL of each strain and grown at 37 °C for 48 h. To assess radial growth, GMM plates supplemented with DMSO, 0.125, 0.25, 0.5, 1, or 8 μg/mL caspofungin were point inoculated with $10^4$ spores in triplicate for each strain and grown at 37 °C for 4 days before measuring colony diameter.

## Statistics and reproducibility

Micrographs are representative of at least three independent experimental replications. Northern blots are representative of at least two experimental replications producing similar results. Quantitative analyses of oxylipins were conducted at least twice to confirm reproducibility of findings in independent experiments. All germling lysis experiments were repeated at least two independent times to confirm experimental reproducibility of statistical trends.

## Reporting summary

Further information on research design is available in the Nature Portfolio Reporting Summary linked to this article.

# Data availability

All relevant methods of data collection and analysis are included in the methods section. The annotated reference genome of *A. fumigatus* Af293 was obtained from FungiDB (release 52) [https://fungidb.org/fungidb/app/]. The RNA sequencing data supporting the findings in this study have been deposited to the NCBI Gene Expression Omnibus with the identifier GSE231238. The list of differentially expressed genes from the RNA sequencing experiment is provided in Supplementary Data 1. All other data obtained to support the findings of this study are available within the article and supplementary materials or from the corresponding author upon request. Source data are provided with this paper.

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

## Acknowledgements

This study was funded in part by the National Institutes of Health R01 AI50669-03 and R01 GM112739-08 to N.P.K. We thank the Fundação de Amparo à Pesquisa do Estado de São Paulo (FAPESP) grant numbers 2021/04977-5 (G.H.G.), and the Conselho Nacional de Desenvolvimento Científico e Tecnológico (CNPq), FAPESP and Fundação Coordenação de Aperfeiçoamento do Pessoal do Ensino Superior (CAPES) grant number 405934/2022-0 (The National Institute of Science and Technology INCT Funvir), and CNPq 301058/2019-9 from Brazil to G.H.G. and the National Institutes of Health/National Institute of Allergy and Infectious Diseases (R01AI153356) from the USA (G.H.G.). This work was also funded by the Joint Canada-Israel Health Research Program, jointly supported by the Azrieli Foundation, Canada's International Development Research Centre, Canadian Institutes of Health Research, and the Israel Science Foundation (G.H.G.).

## Author contributions

D.G.C., G.H.G. and N.P.K. developed the study and designed the experiments. D.G.C., S.C.P. and J.W.B. performed the research. D.G.C., S.C.P., G.H.G. and N.P.K. analyzed and interpreted the data. D.G.C. and N.P.K. wrote the paper. N.P.K. provided supplies for the study. All authors reviewed the paper. N.P.K. is the corresponding author for the paper.

## Competing interests

The authors declare no competing interests.
