## [Peer Review File · Nature Communications]

An oxylipin signal confers protection against antifungal echinocandins in pathogenic aspergilliReviewer #1 (Remarks to the Author):

This manuscript describes the observation that an oxylipin synthesized by *A. fumigatus* modulates the activity of echinocandins toward the fungus. Specifically, it suppresses the tip lysis facet of their mode of action. They show that echinocandin treatment induces the gene involved in its biosynthesis and that echinocandins induce the production of the oxylipin by isolating the lipid from treated and untreated cultures. The transcription factor ZpfA which was known to mediate the effects of this oxylipin on cell wall gene expression was also shown to be required for the suppression.

The premise of the work was based on the observation that oxylipin and echinocandins have shared effects on gene expression and branching phenotypes. So in some ways this is a synthesis of previous studies and they use two data sets from previous work to make this point.

Although this is certainly an interesting observation, the impact of the paper would be dramatically improved if these observational data were bolstered by mechanism-directed experiments. The observation of overlapping DEG sets between caspo and the oxylipin are not followed up. For example, what is the transcriptional profile of cells treated with both caspo and the oxylipin? What oxylipin induced genes are dependent on ZpfA and are there downstream targets that could emerge. Does oxylipin protect against other cell wall active agents? Does it rescue the defects of cell wall mutants or the susceptibility of cell wall mutants? Does it alter cell wall composition?

At this point, the data presented provide a very interesting observation that could lead to a very interesting and impactful piece of work once more complete and detailed mechanistic information are generated.

Additional suggestions

1. A more complete description of the microscopy based lysis assay would help the reader understand the data better---ie that it is a time based assay and information on how many cells and data points go into the analysis. How many cells are included in the analysis and how are the data processed to yield the survival rates?
2. For the lysis/survival data in the figures, three data points are listed. What do those represent? From the methods section, I assume those are lysis rates from independent experiments but it is not clear. If that is the case, how many cells were evaluated and are those data normally distributed?
3. Similar question for the branching data.
4. In some places one way anova is used and in others two way anova is used. What is the rationale for the change.
5. The different strains were treated with different concentrations of echinocandin in Fig. 4; why?
6. There is a significant difference in the amount of lysis observed for the two lab strains. Do they secrete different amounts of oxylipin?

Reviewer #2 (Remarks to the Author):

This manuscript extends previous observations from the senior author that documented the ability of oxylipins (i.e., 5,8-diHODE) to regulate aspects of hyphal morphogenesis such as branching in *Aspergillus fumigatus*. Based on the striking similarity between this response and that elicited by treatment with echinocandins, the authors focus this study on the role of oxylipins in the effects caused by treatment with caspofungin and micafungin. The authors first use transcriptome analysis to demonstrate significant overlap in the gene expression responses to 5,8-diHODE and caspofungin. They then show that caspofungin treatment induces production of 5,8-diHODE (and 8-HODE) via PpoA. They follow this by showing that co-treatment of 5,8-diHODE with caspofungin provides a protective effect in term of limiting tip lysis but not other features of the morphological response to caspofungin. The authors demonstrate that this effect is shared with other isolates of *A. fumigatus* as well as other *Aspergillus* species, and that it also works against micafungin. Lastly, the authors provide evidence that the transcription factor ZpfA, which mediates the morphological

response to 5,8-diHODE, is partially required for the protective effects of oxylipins.

The results presented in this manuscript are truly novel and exciting. The link between oxylipins and the response to echinocandins is somewhat unexpected, and has potential clinical implications for the treatment of a notable fungal pathogen. At the same time, the manuscript would benefit from additional evidence that solidifies the functional and clinical implications of the results. On the functional side, the data suggest that caspofungin-mediated cell wall damage somehow activates PpoA to trigger 5,8-diHODE production, which then acts via ZfpA to induce expression of genes that limit tip lysis. According to this model, the absence of ppoA (perhaps in combination with loss of ppoC) should result in hypersensitivity to caspofungin. Also, the model would predict that the cell wall integrity response as mediated by MpkA would be needed for expression of PpoA. On the clinical side, the authors do not provide any evidence that 5,8-diHODE is protective in an animal (or waxmoth larvae) model, or that a ppoA mutant increases the effectiveness of caspofungin in such a model.

Other points;

lines 98-99. No data are provided that document the effect of these treatments.

Reviewer #3 (Remarks to the Author):

This is very interesting story of new findings in which treatment of *A. fumigatus* with caspofungin or micafungin, but not voriconazole, induces production of 5,8-diHODE by the fungal oxygenase PpoA. Treatment with this fungal oxylipin blocks echinocandin mediated lysis of hyphal tips, requiring the transcription factor ZfpA for full protection. Actually, this finding is a followed-up study in the same lab in 2020, Niu, M. et al. has reported that "Fungal oxylipins direct programmed developmental switches in filamentous fungi. *Nat. Commun.* 11, 5158 (2020). In this study, they found diverse isolates of *A. fumigatus* and two other 5,8-diHODE producing *Aspergillus* pathogens could response to 5,8-diHODE through requiring the transcription factor ZfpA. Importance for this finding may indicate further studies into the genes regulated by 5,8-diHODE and ZfpA as potential targets to enhance antifungal efficacy by unlocking fungicidal activity of echinocandin antifungals. Thus, I think findings are important and novel, writing is logic and data are able to support conclusions. The methods data analysis are sound.

1. PpoA oxylipins protect *Aspergillus* against the fungicidal activity of caspofungin

How about inhibitors of PpoA oxylipins could affect activity of caspofungin?

2. How about relationship between ZfpA with PpoA

3. ZfpA is a partial mediator of 5,8-diHODE protection against echinocandin antifungals.

Do the conserved C2H2 site-directed domain mutant of transcription factor ZfpA could also affect PpoA expression and then resulted in susceptible to caspofungin

4. Does inhibitors of 5,8-diHODE could be as potential novel-type antifungals?

5. In section of Method: it is not clear how to measure "Percent survival of *Aspergillus* germlings.

6. A working model should be good to be understood for this story

Minor : Figure 2 (B,C) 5,8-diHODE and 8-HODE per milligram of dry biomass extracted from fungal tissue (B) or culture supernatant (C) should be Figure 2 (C,D)

Figure 2B should labeled as WT CEA10

REVIEWER COMMENTS

Reviewer #1 (Remarks to the Author):

This manuscript describes the observation that an oxylipin synthesized by *A. fumigatus* modulates the activity of echinocandins toward the fungus. Specifically, it suppresses the tip lysis facet of their mode of action. They show that echinocandin treatment induces the gene involved in its biosynthesis and that echinocandins induce the production of the oxylipin by isolating the lipid from treated and untreated cultures. The transcription factor ZpfA which was known to mediate the effects of this oxylipin on cell wall gene expression was also shown to be required for the suppression.

The premise of the work was based on the observation that oxylipin and echinocandins have shared effects on gene expression and branching phenotypes. So in some ways this is a synthesis of previous studies and they use two data sets from previous work to make this point.

Although this is certainly an interesting observation, the impact of the paper would be dramatically improved if these observational data were bolstered by mechanism-directed experiments. The observation of overlapping DEG sets between caspo and the oxylipin are not followed up. For example, what is the transcriptional profile of cells treated with both caspo and the oxylipin? What oxylipin induced genes are dependent on ZpfA and are there downstream targets that could emerge.

Response: Thank you for your observations and suggestions. Ultimately, we felt that the secondary analysis of these published RNA-seq data sets did not contribute significantly to our study. Instead, we thought it best to perform our own RNA-sequencing experiment of the $\Delta zfpA$ and OE::*zfpA* strains as the overexpression ZfpA strain phenocopies 5,8-diHODE treatment including its resistance to caspofungin mediated tip lysis (New Supplementary Data 1). Our analysis of DEGs of the $\Delta zfpA$ and OE::*zfpA* revealed altered regulation of many chitin metabolic genes which we posited may be responsible for 5,8-diHODE protection against caspofungin tip lysis described in (New Fig. 6A). We followed this with cell wall chitin analysis by CFW staining and found that the cotreatment of caspofungin and 5,8-diHODE induces higher levels of chitin in the fungus than either alone, supporting a cell wall mechanism of reinforcement (New Fig. 6B). These additions are described in lines 275-308 of the results and 424-434 of the discussion and illustrated in our model figure (New Fig. 7). As we will discuss more below, analysis of the DEGs implicated the MpkA pathway which we show to have partial involvement in protection by 5,8-diHODE as described in lines 295-308 and 435-458 (New Fig. 6C).

Does oxylipin protect against other cell wall active agents?

Response: We were similarly curious as to whether this oxylipin was protective against other cell wall active agents. To address this question, we asked if 5,8-diHODE could protect against congo red and found that it could not as described in lines 198-203 and 370-372 (New Fig. 3D). We believe 5,8-diHODE is a specific response to a weakening of FKS activity and possibly the induction is associated with specific cell wall fragments, which we hope to test in the future. Such an undertaking will take considerable time and effort and be the basis for an entirely new research focus of the lab.

Does it rescue the defects of cell wall mutants or the susceptibility of cell wall mutants?

Response: Are you asking if 5,8-diHODE rescues cell wall mutants? Yes, it does. For example, the *zfpA* deletion strain is a cell wall mutant (little chitin) and 5,8-diHODE renders this mutant more resistant to caspofungin, now tested in three *Aspergillus* spp. (New Fig. 5E-F). We further assessed this question by including the $\Delta mpkA$ mutant. We found that 5,8-diHODE conferred statistical protection to the $\Delta mpkA$ mutant but it still remained highly susceptible to caspofungin tip lysis (New Fig. 6C). We also looked at a $\Delta crzA$ mutant which has been identified to impact cell wall composition where saw protection from 5,8-diHODE comparable to the WT strain (New Fig. 6D). These new findings are described in lines 295-308 and 435-458.

Does it alter cell wall composition?

Response: We know from our prior work that 5,8-diHODE increases hyphal chitin detectable by CFW staining (Niu et al. 2020), and in this study we have extended this finding to 8-HODE as well (Supplementary Fig. 2). Further, we found that the combination of 5,8-diHODE and caspofungin increased chitin more than each alone as described in lines 287-294 and 427-434 (New Fig. 6B)

At this point, the data presented provide a very interesting observation that could lead to a very interesting and impactful piece of work once more complete and detailed mechanistic information are generated.

Response: Our new data implicates chitin as the reinforcing agent as described in lines 294-308 and 424-434 (New Fig. 6A-B). As mentioned above, our data is leading to an involvement of cell wall fragments which will surely be a challenging but exciting new direction in our lab. This cannot be solved in 1 year. However, to strengthen why this will be a key area to focus on, we have now asked if the oxylipin response is conserved in other fungi. We found that caspofungin induces *ppoA* in two other *Aspergillus* spp., *A. nidulans* and *A. flavus*, and that 5,8-diHODE protects both species from caspofungin tip lysis described in lines 214-224 and 381-386 (New Fig. 4A-B and Fig. 4C-D). Further, we created *zfpA* deletion and overexpression mutants in both species and found that ZfpA contributes to caspofungin tolerance in both described in lines 248-265 and 411-420 (New Fig. 5E-F). This strengthens our contention that the 5,8-diHODE signaling pathway is conserved, at least among the aspergilli and we speculate beyond this genus. It is interesting to consider that this pathway may have evolved from interactions of soil fungi with native echinocandin producing microbes.

Additional suggestions

1. A more complete description of the microscopy based lysis assay would help the reader understand the data better---ie that it is a time based assay and information on how many cells and data points go into the analysis. How many cells are included in the analysis and how are the data processed to yield the survival rates?

Response: To help provide the reader a better understanding, we elaborated on the details of this assay in the lines 592-605 of the methods section.

2. For the lysis/survival data in the figures, three data points are listed. What do those represent? From the methods section, I assume those are lysis rates from independent experiments but it is not clear. If that is the case, how many cells were evaluated and are those data normally distributed?

Response: In addition to providing a more detailed description of the methods (lines 592-605), we added the following clarification to the legends of all figures containing lysis experiments: "Data points each represent percent survival of 99 germlings assessed in independent wells."

3. Similar question for the branching data.

Response: As previously published by our lab (Niu et al. 2020 *Nature Communications*), the data points represent eight individual hyphae per condition in a single experiment. We have added this detail to the legends of Supplementary Fig. 1 and 7.

4. In some places one-way anova is used and in others two way anova is used. What is the rationale for the change.

Response: In experiments where conditions were defined by only one factor (i.e. treatment type), one-way ANOVA was used. Two-way ANOVA was used in experiments where conditions were defined by two factors (i.e. treatment and genotype).

5. The different strains were treated with different concentrations of echinocandin in Fig. 4; why?

Response: For caspofungin, we selected the lowest concentrations that yielded consistent rates of caspofungin tip lysis in our hands for each strain to avoid any paradoxical effects of higher concentrations. Micafungin is known to be effective at lower concentrations than caspofungin, so initially we reduced proportionally the concentrations used relative to those of caspofungin for each strain. However, we have since repeated this experiment using the same concentrations that were used for caspofungin and replaced Fig. 3B-C to show the new data.

6. There is a significant difference in the amount of lysis observed for the two lab strains. Do they secrete different amounts of oxylipin?

Response: Yes, the two strains produce different amounts of 5,8 with Af293 producing more (Almaliki et al. 2023 *J of Fungi*). Also, the addition of 0.01% w/v yeast extract to GMM media in experiments with Af293 as explained in lines 477-480 may partially explain the observed difference between the Af293 and CEA10 strains. However, we did not conduct any experiments directly comparing lysis between the two, so we cannot draw statistical conclusions about

differences in lysis between these two isolates. Nonetheless, when the lysis experiments with micafungin were repeated using the same concentrations as for caspofungin, we observed more similar rates of lysis in the two strains (New Fig. 3B-C). The key observation is that both strains are protected in addition to *A. flavus* and *A. nidulans*.

Reviewer #2 (Remarks to the Author):

This manuscript extends previous observations from the senior author that documented the ability of oxylipins (i.e., 5,8-diHODE) to regulate aspects of hyphal morphogenesis such as branching in *Aspergillus fumigatus*. Based on the striking similarity between this response and that elicited by treatment with echinocandins, the authors focus this study on the role of oxylipins in the effects caused by treatment with caspofungin and micafungin. The authors first use transcriptome analysis to demonstrate significant overlap in the gene expression responses to 5,8-diHODE and caspofungin. They then show that caspofungin treatment induces production of 5,8-diHODE (and 8-HODE) via PpoA. They follow this by showing that co-treatment of 5,8-diHODE with caspofungin provides a protective effect in term of limiting tip lysis but not other features of the morphological response to caspofungin. The authors demonstrate that this effect is shared with other isolates of *A. fumigatus* as well as other *Aspergillus* species, and that it also works against micafungin. Lastly, the authors provide evidence that the transcription factor ZpfA, which mediates the morphological response to 5,8-diHODE, is partially required for the protective effects of oxylipins.

The results presented in this manuscript are truly novel and exciting. The link between oxylipins and the response to echinocandins is somewhat unexpected, and has potential clinical implications for the treatment of a notable fungal pathogen. At the same time, the manuscript would benefit from additional evidence that solidifies the functional and clinical implications of the results. On the functional side, the data suggest that caspofungin-mediated cell wall damage somehow activates PpoA to trigger 5,8-diHODE production, which then acts via ZpfA to induce expression of genes that limit tip lysis.

Response: Thank you for your kind words.

According to this model, the absence of ppoA (perhaps in combination with loss of ppoC) should result in hypersensitivity to caspofungin.

Response: We also expected the $\Delta ppoA$ mutant might show increased susceptibility to caspofungin mediated tip lysis. Although we were not able to detect a difference in germling lysis between the $\Delta ppoA$ and WT strains, we did find that the OE::ppoA strain showed reduced lysis compared to the wild type supporting the protective role of 5,8-diHODE in the echinocandin response. We have added this new data to the results section in lines 140-146 (New Supplementary Fig. 4). We do not currently suspect a detectable role for PpoC in protection against caspofungin given that there was no detectable increase in the production of its oxylipin product 10-HODE under treatment with caspofungin (New Supplementary Fig. 3).

Also, the model would predict that the cell wall integrity response as mediated by MpkA would be needed for expression of PpoA.

Response: We too were interested in the mechanism by which caspofungin induces ppoA expression. Interestingly, RNA sequencing of the $\Delta zfpA$ and OE::zfpA mutants revealed that ppoA expression is regulated by ZfpA (New Supplementary Data 1). Furthermore, Northern blot analysis of $\Delta zfpA$ mutants in Af293 and CEA10 backgrounds showed that ZfpA is required for the robust induction of ppoA by caspofungin as described in lines 238-247 of the results and in lines 402-410 of the discussion (New Fig. 5C-D). Indeed, RNA-sequencing of $\Delta zfpA$ and OE::zfpA mutants also indicated the possible involvement of MpkA, and we found that 5,8-diHODE could not fully protect a $\Delta mpkA$ mutant against caspofungin mediated tip lysis suggesting it is involved as described in lines 295-308 and 435-458 (New Fig. 6C). Our new model posits that caspofungin activation of ZfpA turns on production of 5,8-diHODE by PpoA which further activates ZfpA in a manner of positive feedback in addition to other yet identified transcription factors involved in protection against echinocandin tip lysis. We illustrate this new model of a protective ZfpA/PpoA signaling relay in New Fig. 7.

On the clinical side, the authors do not provide any evidence that 5,8-diHODE is protective in an animal (or waxmoth larvae) model, or that a ppoA mutant increases the effectiveness of caspofungin in such a model.

Response: We are similarly interested in how oxylipins may impact antifungal susceptibility during infection. However, we note that oxylipins similar to those produced in aspergilli also play functional roles in animal physiology including in immune signaling. Therefore, we feel that unintended effects of 5,8-diHODE on host physiology could obfuscate its

impact on antifungal sensitivity *in vivo*. In fact, we have recently published that *Aspergillus* oxylipins show potential agonist and antagonist activity on the mammalian GPCR G2A. (Steffan et al. 2023 *Frontiers in Immunology*). Still, this is something to consider down the road if we can procure enough 5,8-diHODE for animal tests. Along these lines, however, we have previously shown that the oxylipin responsive transcription factor ZfpA contributes to antifungal sensitivity in an *in vivo* zebrafish model (Schoen et al. 2023 *PLOS Pathogens*). With the current study, we focused on fungal physiology from a more ecological perspective. Following from this, we expanded on our investigation into the conservation of this protective oxylipin signal in the related species *A. flavus* and *A. nidulans*. We found that *A.fl.ppoA* and *A.n.ppoA* were both induced by caspofungin treatment in *A. flavus* and *A. nidulans* respectively described in lines 214-216 and lines 381-382 (new Fig. 4A-B). We also constructed deletion and overexpression mutants of the ZfpA orthologues in each species in order to assess whether the function of this transcription factor was conserved. Our findings suggest that the protective ZfpA/PpoA signaling relay is conserved in *A. flavus* and *A. nidulans*, but likely protection by 5,8-diHODE against echinocandin lysis involves multiple transcription factors in these species as described in lines 248-273 and 411-423 (new Fig. 5C-F).

Other points;

lines 98-99. No data are provided that document the effect of these treatments.

Response: These phenotypes in response to 5,8-diHODE and caspofungin have been previously published (Niu et al. 2020 *Nature Communications*; Moreno-Velásquez et al. 2017 *Antimicrob Agents Chemother.*), but we have recapitulated the prior studies and added the appropriate data showing the same effect of 8-HODE (New Supplementary Fig. 1 and 2).

Reviewer #3 (Remarks to the Author):

This is very interesting story of new findings in which treatment of *A. fumigatus* with caspofungin or micafungin, but not voriconazole, induces production of 5,8-diHODE by the fungal oxygenase PpoA. Treatment with this fungal oxylipin blocks echinocandin mediated lysis of hyphal tips, requiring the transcription factor ZfpA for full protection. Actually, this finding is a followed-up study in the same lab in 2020, Niu, M. et al. has reported that "Fungal oxylipins direct programmed developmental switches in filamentous fungi. *Nat. Commun.* 11, 5158 (2020). In this study, they found diverse isolates of *A. fumigatus* and two other 5,8-diHODE producing *Aspergillus* pathogens could response to 5,8-diHODE through requiring the transcription factor ZfpA. Importance for this finding may indicate further studies into the genes regulated by 5,8-diHODE and ZfpA as potential targets to enhance antifungal efficacy by unlocking fungicidal activity of echinocandin antifungals. Thus, I think findings are important and novel, writing is logic and data are able to support conclusions. The methods data analysis are sound.

1. PpoA oxylipins protect *Aspergillus* against the fungicidal activity of caspofungin
How about inhibitors of PpoA oxylipins could affect activity of caspofungin?

Response: We are not currently aware of any PpoA specific inhibitors, but this is something that we would be very interested to know ourselves.

2. How about the relationship between ZfpA with PpoA

Response: This is a good question. Additional work has revealed a positive feedback relay between ZfpA and PpoA conserved in all three *Aspergillus* spp. tested, as described in lines 238-273 of the results and in lines 402-423 of the discussion (New Fig. 6B-F). We present a model of how this relay leads to protection from lysis in new model Fig. 7.

3. ZfpA is a partial mediator of 5,8-diHODE protection against echinocandin antifungals.

Do the conserved C2H2 site-directed domain mutant of transcription factor ZfpA could also affect PpoA expression and then resulted in susceptible to caspofungin

Response: Although we did not construct any ZfpA C2H2 site directed mutants, we believe our New Fig. 5 discussed in the above response answers this question in that ZfpA does impact the expression of PpoA and is needed for full *ppoA* expression upon caspofungin treatment.

4. Does inhibitors of 5,8-diHODE could be as potential novel-type antifungals?

Response: We have not as of yet identified any 5,8-diHODE inhibitors. We do have the *ppoA* deletion mutant but possibly the reviewer is wondering about 5,8-diHODE antagonists? This would be an interesting avenue to explore in the future.

5. In section of Method: it is not clear how to measure "Percent survival of *Aspergillus* germlings.

Response: To help provide the reader a better understanding of this assay, we elaborated on the details of this assay in methods section lines 592-605.

6. A working model should be good to be understood for this story

Response: We appreciate this recommendation and have added a final figure with a working model (New Fig. 7).

Minor : Figure 2 (B,C) 5,8-diHODE and 8-HODE per milligram of dry biomass extracted from fungal tissue (B) or culture supernatant (C) should be Figure 2 (C,D)
Figure 2B should be labeled as WT CEA10

Response: We have corrected these errors and thank the reviewer for bringing them to our attention.

Reviewer #1 (Remarks to the Author):

The authors have addressed my suggestions.

Reviewer #2 (Remarks to the Author):

The authors have done a terrific job of responding to my comments and those of the other reviewers. The additional data provided in the revised version of this manuscript have substantially improved it and support a reasonable mechanistic hypothesis that accounts for the protective effects of oxylipins upon echinocandin exposure. Accordingly, I fully support publication.

My only comment refers to Fig. 6B. It would be helpful to provide a supplemental image of the Calcofluor stained hyphae in support of the graph. Also, it would potentially reveal whether the additional chitin was uniformly distributed or was concentrated at specific sites (e.g., tips, septa).

REVIEWERS' COMMENTS

Reviewer #1 (Remarks to the Author):

The authors have addressed my suggestions.

Response: We thank the reviewer for their remarks.

Reviewer #2 (Remarks to the Author):

The authors have done a terrific job of responding to my comments and those of the other reviewers. The additional data provided in the revised version of this manuscript have substantially improved it and support a reasonable mechanistic hypothesis that accounts for the protective effects of oxylipins upon echinocandin exposure. Accordingly, I fully support publication.

Response: We thank the reviewer for their kind remarks.

My only comment refers to Fig. 6B. It would be helpful to provide a supplemental image of the Calcofluor stained hyphae in support of the graph. Also, it would potentially reveal whether the additional chitin was uniformly distributed or was concentrated at specific sites (e.g., tips, septa).

Response: We thank the reviewer for their valuable comment. Unfortunately, the differences here are not easily discernable by the naked eye between the caspofungin treated and cotreated hyphae. However, we are also very interested in how the distribution of chitin in the hyphae may differ between conditions and are working to develop an empirical method to quantitatively assess this. Thus, we hesitate to introduce any speculation in this study but hope to build a mechanistic understanding of this chitin induction in future work.